# Dichotomize and Generalize: PAC-Bayesian Binary Activated Deep Neural Networks

**Gaël Letarte**
Université Laval
Canada
gael.letarte.1@ulaval.ca

**Pascal Germain**
Inria
France
pascal.germain@inria.fr

**Benjamin Guedj**
Inria and University College London
France and United Kingdom
benjamin.guedj@inria.fr

**François Laviolette**
Université Laval
Canada
francois.laviolette@ift.ulaval.ca

## Abstract

We present a comprehensive study of multilayer neural networks with binary activation, relying on the PAC-Bayesian theory. Our contributions are twofold: (i) we develop an end-to-end framework to train a binary activated deep neural network, (ii) we provide nonvacuous PAC-Bayesian generalization bounds for binary activated deep neural networks. Our results are obtained by minimizing the expected loss of an architecture-dependent aggregation of binary activated deep neural networks. Our analysis inherently overcomes the fact that binary activation function is non-differentiable. The performance of our approach is assessed on a thorough numerical experiment protocol on real-life datasets.

## 1   Introduction

The remarkable practical successes of deep learning make the need for better theoretical understanding all the more pressing. The PAC-Bayesian theory has recently emerged as a fruitful framework to analyze generalization abilities of deep neural network. Inspired by precursor work of Langford and Caruana [2001], nonvacuous risk bounds for multilayer architectures have been obtained by Dziugaite and Roy [2017], Zhou et al. [2019]. Although informative, these results do not explicitly take into account the network architecture (number of layers, neurons per layer, type of activation function). A notable exception is the work of Neyshabur et al. [2018] which provides a PAC-Bayesian analysis relying on the network architecture and the choice of ReLU activation function. The latter bound arguably gives insights on the generalization mechanism of neural networks (namely in terms of the spectral norms of the learned weight matrices), but their validity hold for some margin assumptions, and they are likely to be numerically vacuous.

We focus our study on deep neural networks with a sign activation function. We call such networks *binary activated multilayer* (BAM) networks. This specialization leads to nonvacuous generalization bounds which hold under the sole assumption that training samples are *iid*. We provide a PAC-Bayesian bound holding on the generalization error of a continuous aggregation of BAM networks. This leads to an original approach to train BAM networks, named PBGNet. The building block of PBGNet arises from the specialization of PAC-Bayesian bounds to linear classifiers [Germain et al., 2009], that we adapt to deep neural networks. The term *binary neural networks* has been coined by Bengio [2009], and further studied in Hubara et al. [2016, 2017], Soudry et al. [2014]: it refers to neural networks for which both the activation functions and the weights are binarized (in contrast

with BAM networks). These architectures are motivated by the desire to reduce the computation and memory footprints of neural networks.

Our theory-driven approach is validated on real life datasets, showing competitive accuracy with $\tt tanh$-activated multilayer networks, and providing nonvacuous generalization bounds.

**Organisation of the paper.** We formalize our framework and notation in Section 2, along with a presentation of the PAC-Bayes framework and its specialization to linear classifiers. Section 3 illustrates the key ideas we develop in the present paper, on the simple case of a two-layers neural network. This is then generalized to deep neural networks in Section 4. We present our main theoretical result in Section 5: a PAC-Bayesian generalization bound for binary activated deep neural networks, and the associated learning algorithm. Section 6 presents the numerical experiment protocol and results. The paper closes with avenues for future work in Section 7.

## 2  Framework and notation

We stand in the supervised binary classification setting: given a real input vector[1] $\mathbf{x} \in \mathbb{R}^{d_0}$, one wants to predict a label $y \in \{-1, 1\}$. Let us consider a neural network of $L$ *fully connected* layers with a (binary) sign activation function: $\mathrm{sgn}(a) = 1$ if $a > 0$ and $\mathrm{sgn}(a) = -1$ otherwise.[2] We let $d_k$ denote the number of neurons of the $k^{\mathrm{th}}$ layer, for $k \in \{1, \ldots, L\}$; $d_0$ is the input data point dimension, and $D := \sum_{k=1}^{L} d_{k-1} d_k$ is the total number of parameters. The output of the (deterministic) BAM network on an input data point $\mathbf{x} \in \mathbb{R}^{d_0}$ is given by

$$f_\theta(\mathbf{x}) = \mathrm{sgn}\big(\mathbf{W}_L \mathrm{sgn}\big(\mathbf{W}_{L-1} \mathrm{sgn}\big(\ldots \mathrm{sgn}\big(\mathbf{W}_1 \mathbf{x}\big)\big)\big)\big), \tag{1}$$

where $\mathbf{W}_k \in \mathbb{R}^{d_k \times d_{k-1}}$ denotes the weight matrices. The network is thus parametrized by $\theta = \mathrm{vec}\big(\{\mathbf{W}_k\}_{k=1}^L\big) \in \mathbb{R}^D$. The $i^{\mathrm{th}}$ line of matrix $\mathbf{W}_k$ will be denoted $\mathbf{w}_k^i$. For binary classification, the BAM network final layer $\mathbf{W}_L \in \mathbb{R}^{1 \times d_{L-1}}$ has one line ($d_L = 1$), that is a vector $\mathbf{w}_L \in \mathbb{R}^{d_{L-1}}$, and $f_\theta : \mathbb{R}^{d_0} \to \{-1, 1\}$.

### 2.1  Elements from the PAC-Bayesian theory

The Probably Approximately Correct (PAC) framework [introduced by Valiant, 1984] holds under the frequentist assumption that data is sampled in an *iid* fashion from a data distribution $\mathcal{D}$ over the input-output space. The learning algorithm observes a finite training sample $S = \{(\mathbf{x}_i, y_i)\}_{i=1}^n \sim \mathcal{D}^n$ and outputs a predictor $f : \mathbb{R}^{d_0} \to [-1, 1]$. Given a loss function $\ell : [-1, 1]^2 \to [0, 1]$, we define $\mathcal{L}_\mathcal{D}(f)$ as the generalization loss on the data generating distribution $\mathcal{D}$, and $\widehat{\mathcal{L}}_S(f)$ as the empirical error on the training set, given by

$$\mathcal{L}_\mathcal{D}(f) = \mathop{\mathbf{E}}_{(\mathbf{x}, y) \sim \mathcal{D}} \ell(f(\mathbf{x}), y), \quad \text{and} \quad \widehat{\mathcal{L}}_S(f) = \frac{1}{n} \sum_{i=1}^n \ell(f(\mathbf{x}_i), y_i).$$

PAC-Bayes considers the expected loss of an aggregation of predictors: considering a distribution $Q$ (called the *posterior*) over a family of predictors $\mathcal{F}$, one obtains PAC upper bounds on $\mathbf{E}_{f \sim Q} \mathcal{L}_\mathcal{D}(f)$. Our work focuses on the linear loss $\ell(y', y) := \frac{1}{2}(1 - yy')$, for which the aggregated loss is equivalent to the loss of the predictor $F_Q(\mathbf{x}) := \mathbf{E}_{f \sim Q} f(\mathbf{x})$, performing a $Q$-aggregation of all predictors in $\mathcal{F}$. In other words, we may upper bound with an arbitrarily high probability the generalization loss $\mathcal{L}_\mathcal{D}(F_Q) = \mathbf{E}_{f \sim Q} \mathcal{L}_\mathcal{D}(f)$, by its empirical counterpart $\widehat{\mathcal{L}}_S(F_Q) = \mathbf{E}_{f \sim Q} \widehat{\mathcal{L}}_S(f)$ and a complexity term, the Kullback-Leibler divergence between $Q$ and a reference measure $P$ (called the *prior* distribution) chosen independently of the training set $S$, given by $\mathrm{KL}(Q\|P) := \int \ln \frac{Q(\theta)}{P(\theta)} Q(\mathrm{d}\theta)$. Since the seminal works of Shawe-Taylor and Williamson [1997], McAllester [1999, 2003] and Catoni [2003, 2004, 2007], the celebrated PAC-Bayesian theorem has been declined in many forms [see Guedj, 2019, for a survey]. The following Theorems 1 and 2 will be useful in the sequel.

**Theorem 1** (Seeger [2002], Maurer [2004]). *Given a prior $P$ on $\mathcal{F}$, with probability at least $1 - \delta$ over $S \sim \mathcal{D}^n$,*

$$\text{for all } Q \text{ on } \mathcal{F}: \quad \text{kl}\left(\widehat{\mathcal{L}}_S(F_Q)\big\|\mathcal{L}_{\mathcal{D}}(F_Q)\right) \leq \frac{\text{KL}(Q\|P) + \ln\frac{2\sqrt{n}}{\delta}}{n}, \tag{2}$$

*where $\text{kl}(q\|p) := q\ln\frac{q}{p} + (1-q)\ln\frac{1-q}{1-p}$ is the Kullback-Leibler divergence between Bernoulli distributions with probability of success $p$ and $q$, respectively.*

**Theorem 2** (Catoni [2007]). *Given $P$ on $\mathcal{F}$ and $C > 0$, with probability at least $1 - \delta$ over $S \sim \mathcal{D}^n$,*

$$\text{for all } Q \text{ on } \mathcal{F}: \quad \mathcal{L}_{\mathcal{D}}(F_Q) \leq \frac{1}{1 - e^{-C}}\left(1 - \exp\left(-C\,\widehat{\mathcal{L}}_S(F_Q) - \frac{\text{KL}(Q\|P) + \ln\frac{1}{\delta}}{n}\right)\right). \tag{3}$$

From Theorems 1 and 2, we obtain PAC-Bayesian bounds on the *linear loss* of the $Q$-aggregated predictor $F_Q$. Given our binary classification setting, it is natural to predict a label by taking the sign of $F_Q(\cdot)$. Thus, one may also be interested in the *zero-one loss* $\ell_{01}(y', y) := \mathbb{1}[\text{sgn}(y') \neq y]$; the bounds obtained from Theorems 1 and 2 can be turned into bounds on the *zero-one loss* with an extra 2 multiplicative factor, using the elementary inequality $\ell_{01}(F_Q(\mathbf{x}), y) \leq 2\ell(F_Q(\mathbf{x}), y)$.

## 2.2 Elementary building block: PAC-Bayesian learning of linear classifiers

The PAC-Bayesian specialization to linear classifiers has been proposed by Langford and Shawe-Taylor [2002], and used for providing tight generalization bounds and a model selection criteria [further studied by Ambroladze et al., 2006, Langford, 2005, Parrado-Hernández et al., 2012]. This paved the way to the PAC-Bayesian bound minimization algorithm of Germain et al. [2009], that learns a linear classifier $f_{\mathbf{w}}(\mathbf{x}) := \text{sgn}(\mathbf{w} \cdot \mathbf{x})$, with $\mathbf{w} \in \mathbb{R}^d$. The strategy is to consider a Gaussian posterior $Q_{\mathbf{w}} := \mathcal{N}(\mathbf{w}, I_d)$ and a Gaussian prior $P_{\mathbf{w}_0} := \mathcal{N}(\mathbf{w}_0, I_d)$ over the space of all linear predictors $\mathcal{F}_d := \{f_{\mathbf{v}} | \mathbf{v} \in \mathbb{R}^d\}$ (where $I_d$ denotes the $d \times d$ identity matrix). The posterior is used to define a linear predictor $f_{\mathbf{w}}$ and the prior may have been learned on previously seen data; a common uninformative prior being the null vector $\mathbf{w}_0 = \mathbf{0}$. With such parametrization, $\text{KL}(Q_{\mathbf{w}}\|P_{\mathbf{w}_0}) = \frac{1}{2}\|\mathbf{w} - \mathbf{w}_0\|^2$. Moreover, the $Q_{\mathbf{w}}$-aggregated output can be written in terms of the Gauss error function $\text{erf}(\cdot)$. In Germain et al. [2009], the erf function is introduced as a loss function to be optimized. Here we interpret it as the predictor output, to be in phase with our neural network approach. Likewise, we study the linear loss of an aggregated predictor instead of the *Gibbs risk* of a stochastic classifier. We obtain (explicit calculations are provided in Appendix A.1 for completeness)

$$F_{\mathbf{w}}(\mathbf{x}) := \mathop{\mathbf{E}}_{\mathbf{v} \sim Q_{\mathbf{w}}} f_{\mathbf{v}}(\mathbf{x}) = \text{erf}\left(\frac{\mathbf{w} \cdot \mathbf{x}}{\sqrt{2}\|\mathbf{x}\|}\right), \quad \text{with } \text{erf}(x) := \frac{2}{\sqrt{\pi}}\int_0^x e^{-t^2}dt. \tag{4}$$

Given a training set $S \sim \mathcal{D}^n$, Germain et al. [2009] propose to minimize a PAC-Bayes upper bound on $\mathcal{L}_{\mathcal{D}}(F_{\mathbf{w}})$ by gradient descent on $\mathbf{w}$. This approach is appealing as the bounds are valid uniformly for all $Q_{\mathbf{w}}$ (see Equations 2 and 3). In other words, the algorithm provides both a learned predictor and a generalization guarantee that is rigorously valid (under the *iid* assumption) even when the optimization procedure did not find the global minimum of the cost function (either because it converges to a local minimum, or early stopping is used). Germain et al. [2009] investigate the optimization of several versions of Theorems 1 and 2. The minimization of Theorem 1 generally leads to tighter bound values, but empirical studies show lowest accuracy as the procedure conservatively prevents overfitting. The best empirical results are obtained by minimizing Theorem 2 for a fixed hyperparameter $C$, selected by cross-validation. Minimizing Equation (3) amounts to minimizing

$$C\,n\,\widehat{\mathcal{L}}_S(F_{\mathbf{w}}) + \text{KL}(Q_{\mathbf{w}}\|P_{\mathbf{w}_0}) = C\frac{1}{2}\sum_{i=1}^n \text{erf}\left(-y_i\frac{\mathbf{w} \cdot \mathbf{x}_i}{\sqrt{2}\|\mathbf{x}_i\|}\right) + \frac{1}{2}\|\mathbf{w} - \mathbf{w}_0\|^2. \tag{5}$$

In their discussion, Germain et al. [2009] observe that the objective in Equation (5) is similar to the one optimized by the soft-margin Support Vector Machines [Cortes and Vapnik, 1995], by roughly interpreting the *hinge loss* $\max(0, 1 - yy')$ as a convex surrogate of the *probit loss* $\text{erf}(-yy')$. Likewise, Langford and Shawe-Taylor [2002] present this parameterization of the PAC-Bayes theorem as a margin bound. In the following, we develop an original approach to neural networks based on a slightly different observation: the predictor output given by Equation (4) is reminiscent of the $\tanh$ activation used in classical neural networks (see Figure 3 in the appendix for a visual comparison). Therefore, as the linear *perceptron* is viewed as the *building block* of modern multilayer neural networks, the PAC-Bayesian specialization to binary classifiers is the cornerstone of our theoretical and algorithmic framework for BAM networks.

# 3 The simple case of a one hidden layer network

Let us first consider a network with one hidden layer of size $d_1$. Hence, this network is parameterized by weights $\theta = \text{vec}(\{\mathbf{W}_1, \mathbf{w}_2\})$, with $\mathbf{W}_1 \in \mathbb{R}^{d_1 \times d_0}$ and $\mathbf{w}_2 \in \mathbb{R}^{d_1}$. Given an input $\mathbf{x} \in \mathbb{R}^{d_0}$, the output of the network is

$$f_\theta(\mathbf{x}) = \text{sgn}\big(\mathbf{w}_2 \cdot \text{sgn}(\mathbf{W}_1\mathbf{x})\big). \tag{6}$$

Following Section 2, we consider an isotropic Gaussian posterior distribution centered in $\theta$, denoted $Q_\theta = \mathcal{N}(\theta, I_D)$, over the family of all networks $\mathcal{F}_D = \{f_{\tilde{\theta}} \,|\, \tilde{\theta} \in \mathbb{R}^D\}$. Thus, the prediction of the $Q_\theta$-aggregate predictor is given by $F_\theta(\mathbf{x}) = \mathbf{E}_{\tilde{\theta} \sim Q_\theta} f_{\tilde{\theta}}(\mathbf{x})$. Note that Dziugaite and Roy [2017], Langford and Caruana [2001] also consider Gaussian distributions over neural networks parameters. However, as their analysis is not specific to a particular activation function—experiments are performed with *typical* activation functions (sigmoid, ReLU)—the prediction relies on sampling the parameters according to the posterior. An originality of our approach is that, by studying the sign activation function, we can calculate the exact form of $F_\theta(\mathbf{x})$, as detailed below.

## 3.1 Deterministic network

**Prediction.** To compute the value of $F_\theta(\mathbf{x})$, we first need to decompose the probability of each $\tilde{\theta} = \text{vec}(\{\mathbf{V}_1, \mathbf{v}_2\}) \sim Q_\theta$ as $Q_\theta(\tilde{\theta}) = Q_1(\mathbf{V}_1)Q_2(\mathbf{v}_2)$, with $Q_1 = \mathcal{N}(\mathbf{W}_1, I_{d_0 d_1})$ and $Q_2 = \mathcal{N}(\mathbf{w}_2, I_{d_1})$.

$$F_\theta(\mathbf{x}) = \int_{\mathbb{R}^{d_1 \times d_0}} Q_1(\mathbf{V}_1) \int_{\mathbb{R}^{d_1}} Q_2(\mathbf{v}_2)\text{sgn}(\mathbf{v}_2 \cdot \text{sgn}(\mathbf{V}_1\mathbf{x})) d\mathbf{v}_2 d\mathbf{V}_1$$

$$= \int_{\mathbb{R}^{d_1 \times d_0}} Q_1(\mathbf{V}_1) \, \text{erf}\left(\frac{\mathbf{w}_2 \cdot \text{sgn}(\mathbf{V}_1\mathbf{x})}{\sqrt{2}\|\text{sgn}(\mathbf{V}_1\mathbf{x})\|}\right) d\mathbf{V}_1 \tag{7}$$

$$= \sum_{\mathbf{s} \in \{-1,1\}^{d_1}} \text{erf}\left(\frac{\mathbf{w}_2 \cdot \mathbf{s}}{\sqrt{2d_1}}\right) \int \mathbb{1}[\mathbf{s} = \text{sgn}(\mathbf{V}_1\mathbf{x})]Q_1(\mathbf{V}_1) \, d\mathbf{V}_1 \tag{8}$$

$$= \sum_{\mathbf{s} \in \{-1,1\}^{d_1}} \text{erf}\left(\frac{\mathbf{w}_2 \cdot \mathbf{s}}{\sqrt{2d_1}}\right) \Psi_{\mathbf{s}}(\mathbf{x}, \mathbf{W}_1), \tag{9}$$

where, from $Q_1(\mathbf{V}_1) = \prod_{i=1}^{d_1} Q_1^i(\mathbf{v}_1^i)$ with $Q_1^i := \mathcal{N}(\mathbf{w}_1^i, I_{d_0})$, we obtain

$$\Psi_{\mathbf{s}}(\mathbf{x}, \mathbf{W}_1) := \prod_{i=1}^{d_1} \int_{\mathbb{R}^{d_0}} \mathbb{1}[s_i \, \mathbf{x} \cdot \mathbf{v}_1^i > 0]Q_1^i(\mathbf{v}_1^i) \, d\mathbf{v}_1^i = \prod_{i=1}^{d_1} \underbrace{\left[\frac{1}{2} + \frac{s_i}{2} \text{erf}\left(\frac{\mathbf{w}_1^i \cdot \mathbf{x}}{\sqrt{2}\|\mathbf{x}\|}\right)\right]}_{\psi_{s_i}(\mathbf{x}, \mathbf{w}_1^i)}. \tag{10}$$

Line (7) states that the output neuron is a linear predictor over the hidden layer's activation values $\mathbf{s} = \text{sgn}(\mathbf{V}_1\mathbf{x})$; based on Equation (4), the integral on $\mathbf{v}_2$ becomes $\text{erf}(\mathbf{w}_2 \cdot \mathbf{s}/(\sqrt{2}\|\mathbf{s}\|))$. As a function of $\mathbf{s}$, the latter expression is piecewise constant. Thus, Line (8) discretizes the integral on $\mathbf{V}_1$ as a sum of the $2^{d_1}$ different values of $\mathbf{s} = (s_i)_{i=1}^{d_1}, s_i \in \{-1, 1\}$. Note that $\|\mathbf{s}\|^2 = d_1$.

Finally, one can compute the exact output of $F_\theta(\mathbf{x})$, provided one accepts to compute a sum combinatorial in the number of hidden neurons (Equation 9). We show in forthcoming Section 3.2 that it is possible to circumvent this computational burden and approximate $F_\theta(\mathbf{x})$ by a sampling procedure.

**Derivatives.** Following contemporary approaches in deep neural networks [Goodfellow et al., 2016], we minimize the empirical loss $\widehat{\mathcal{L}}_S(F_\theta)$ by stochastic gradient descent (SGD). This requires to compute the partial derivative of the cost function according to the parameters $\theta$:

$$\frac{\partial \widehat{\mathcal{L}}_S(F_\theta)}{\partial \theta} = \frac{1}{n} \sum_{i=1}^{n} \frac{\partial \ell(F_\theta(\mathbf{x}_i), y_i)}{\partial \theta} = \frac{1}{n} \sum_{i=1}^{n} \frac{\partial F_\theta(\mathbf{x}_i)}{\partial \theta} \, \ell'(F_\theta(\mathbf{x}_i), y_i), \tag{11}$$

with the derivative of the linear loss $\ell'(F_\theta(\mathbf{x}_i), y_i) = -\frac{1}{2}y$.

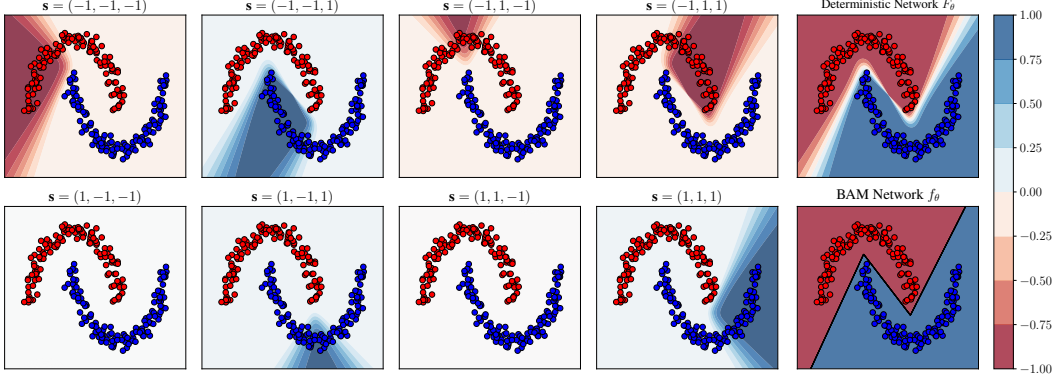

Figure 1: Illustration of the proposed method for a one hidden layer network of size $d_1{=}3$, interpreted as a majority vote over 8 binary representations $\mathbf{s} \in \{-1, 1\}^3$. For each $\mathbf{s}$, a plot shows the values of $F_{\mathbf{w}_2}(\mathbf{s})\Psi_{\mathbf{s}}(\mathbf{x}, \mathbf{W}_1)$. The sum of these values gives the deterministic network output $F_\theta(\mathbf{x})$ (see Eq. 9). We also plot the BAM network output $f_\theta(\mathbf{x})$ for the same parameters $\theta$ (see Eq. 6).

The partial derivatives of the prediction function (Equation 9) according to the hidden layer parameters $\mathbf{w}_1^k \in \{\mathbf{w}_1^1, \dots, \mathbf{w}_1^{d_1}\}$ and the output neuron parameters $\mathbf{w}_2$ are

$$\frac{\partial}{\partial \mathbf{w}_1^k} F_\theta(\mathbf{x}) = \frac{\mathbf{x}}{2^{\frac{3}{2}} \|\mathbf{x}\|} \mathrm{erf}'\left(\frac{\mathbf{w}_1^k \cdot \mathbf{x}}{\sqrt{2}\,\|\mathbf{x}\|}\right) \sum_{\mathbf{s} \in \{-1,1\}^{d_1}} s_k\, \mathrm{erf}\left(\frac{\mathbf{w}_2 \cdot \mathbf{s}}{\sqrt{2d_1}}\right) \left[\frac{\Psi_{\mathbf{s}}(\mathbf{x}, \mathbf{W}_1)}{\psi_{s_k}(\mathbf{x}, \mathbf{w}_1^k)}\right], \tag{12}$$

$$\frac{\partial}{\partial \mathbf{w}_2} F_\theta(\mathbf{x}) = \frac{1}{\sqrt{2d_1}} \sum_{\mathbf{s} \in \{-1,1\}^{d_1}} \mathbf{s}\, \mathrm{erf}'\left(\frac{\mathbf{w}_2 \cdot \mathbf{s}}{\sqrt{2d_1}}\right) \Psi_{\mathbf{s}}(\mathbf{x}, \mathbf{W}_1), \quad \text{with } \mathrm{erf}'(x) := \frac{2}{\sqrt{\pi}} e^{-x^2}. \tag{13}$$

Note that this is an exact computation. A salient fact is that even though we work on non-differentiable BAM networks, we get a structure trainable by (stochastic) gradient descent by aggregating networks.

**Majority vote of learned representations.** Note that $\Psi_{\mathbf{s}}$ (Equation 10) defines a distribution on $\mathbf{s}$. Indeed, $\sum_{\mathbf{s}} \Psi_{\mathbf{s}}(\mathbf{x}, \mathbf{W}_1){=}1$, as $\Psi_{\mathbf{s}}(\mathbf{x}, \mathbf{W}_1) + \Psi_{\bar{\mathbf{s}}}(\mathbf{x}, \mathbf{W}_1) = 2^{-d_1}$ for every $\bar{\mathbf{s}} = -\mathbf{s}$. Thus, by Equation (9) we can interpret $F_\theta$ akin to a majority vote predictor, which performs a convex combination of a linear predictor outputs $F_{\mathbf{w}_2}(\mathbf{s}) := \mathrm{erf}(\mathbf{w}_2 \cdot \mathbf{s}/\sqrt{2d_1})$. The vote aggregates the predictions on the $2^{d_1}$ possible binary representations. Thus, the algorithm does not learn the representations *per se*, but rather the weights $\Psi_{\mathbf{s}}(\mathbf{x}, \mathbf{W}_1)$ associated to every $\mathbf{s}$ given an input $\mathbf{x}$, as illustrated by Figure 1.

### 3.2 Stochastic approximation

Since $\Psi_{\mathbf{s}}$ (Equation 10) defines a distribution, we can interpret the function value as the probability of mapping input $\mathbf{x}$ into the hidden representation $\mathbf{s}$ given the parameters $\mathbf{W}_1$. Using a different formalism, we could write $\Pr(\mathbf{s}|\mathbf{x}, \mathbf{W}_1) = \Psi_{\mathbf{s}}(\mathbf{x}, \mathbf{W}_1)$. This viewpoint suggests a sampling scheme to approximate both the predictor output (Equation 9) and the partial derivatives (Equations 12 and 13), that can be framed as a variant of the REINFORCE algorithm [Williams, 1992] (see the discussion below): We avoid computing the $2^{d_1}$ terms by resorting to a Monte Carlo approximation of the sum. Given an input $\mathbf{x}$ and a sampling size $T$, the procedure goes as follows.

**Prediction.** We generate $T$ random binary vectors $Z := \{\mathbf{s}^t\}_{t=1}^T$ according to the $\Psi_{\mathbf{s}}(\mathbf{x}, \mathbf{W}_1)$-distribution. This can be done by uniformly sampling $z_i^t \in [0, 1]$, and setting $s_i^t = \mathrm{sgn}(\psi_1(\mathbf{x}, \mathbf{w}_1^i) - z_i^t)$. A stochastic approximation of $F_\theta(\mathbf{x})$ is given by $\widehat{F}_\theta(Z) := \frac{1}{T} \sum_{t=1}^T \mathrm{erf}\left(\frac{\mathbf{w}_2 \cdot \mathbf{s}^t}{\sqrt{2d_1}}\right)$.

**Derivatives.** Note that for a given sample $\{\mathbf{s}^t\}_{t=1}^T$, the approximate derivatives according to $\mathbf{w}_2$ (Equation 15 below) can be computed numerically by the automatic differentiation mechanism of deep learning frameworks while evaluating $\widehat{F}_\theta(Z)$ [*e.g.*, Paszke et al., 2017]. However, we need the following Equation (14) to approximate the gradient according to $\mathbf{W}_1$ because $\partial \widehat{F}_\theta(Z)/\partial \mathbf{w}_1^k = 0$.

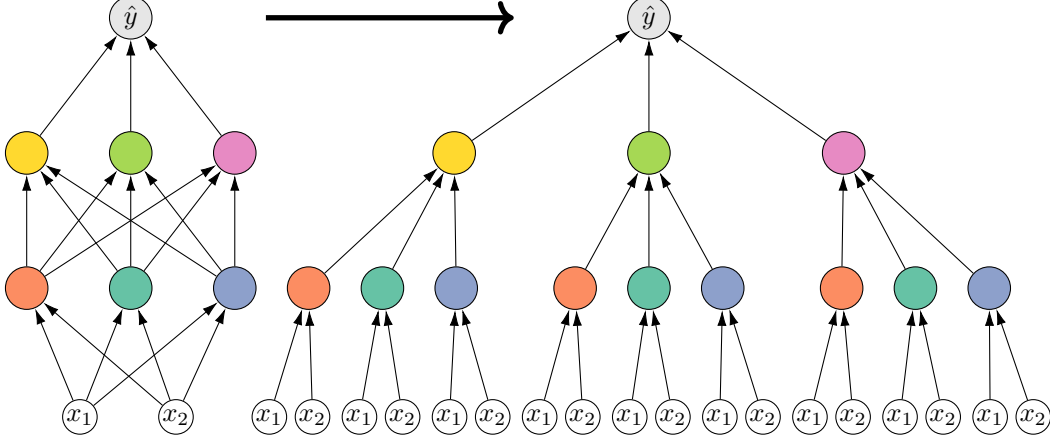

Figure 2: Illustration of the *BAM to tree architecture map* on a three layers network.

$$\frac{\partial}{\partial \mathbf{w}_1^k} F_\theta(\mathbf{x}) \approx \frac{\mathbf{x}}{T \, 2^{\frac{3}{2}} \|\mathbf{x}\|} \mathrm{erf}' \left( \frac{\mathbf{w}_1^k \cdot \mathbf{x}}{\sqrt{2} \|\mathbf{x}\|} \right) \sum_{t=1}^T \frac{s_k^t}{\psi_{s_k^t}(\mathbf{x}, \mathbf{w}_1^k)} \mathrm{erf} \left( \frac{\mathbf{w}_2 \cdot \mathbf{s}^t}{\sqrt{2d_1}} \right) ; \tag{14}$$

$$\frac{\partial}{\partial \mathbf{w}_2} F_\theta(\mathbf{x}) \approx \frac{1}{T \sqrt{2d_1}} \sum_{t=1}^T \mathbf{s}^t \, \mathrm{erf}' \left( \frac{\mathbf{w}_2 \cdot \mathbf{s}^t}{\sqrt{2d_1}} \right) = \frac{\partial}{\partial \mathbf{w}_2} \widehat{F}_\theta(Z) . \tag{15}$$

**Similar approaches to stochastic networks.** Random activation functions are commonly used in generative neural networks, and tools have been developed to train these by gradient descent (see Goodfellow et al. [2016, Section 20.9] for a review). Contrary to these approaches, our analysis differs as the stochastic operations are introduced to estimate a deterministic objective. That being said, Equation (14) can be interpreted as a variant of REINFORCE algorithm [Williams, 1992] to apply the back-propagation method along with discrete activation functions. Interestingly, the formulation we obtain through our PAC-Bayes objective is similar to a commonly used REINFORCE variant [*e.g.*, Bengio et al., 2013, Yin and Zhou, 2019], where the activation function is given by a Bernoulli variable with probability of success $\sigma(a)$, where $a$ is the neuron input, and $\sigma$ is the sigma is the *sigmoid function*. The latter can be interpreted as a surrogate of our $\psi_{s_i}(\mathbf{x}, \mathbf{w}_1^i)$.

## 4 Generalization to multilayer networks

In the following, we extend the strategy introduced in Section 3 to BAM architectures with an arbitrary number of layers $L \in \mathbb{N}^*$ (Equation 1). An apparently straightforward approach to achieve this generalization would have been to consider a Gaussian posterior distribution $\mathcal{N}(\theta, I_D)$ over the BAM family $\{f_{\tilde{\theta}} | \tilde{\theta} \in \mathbb{R}^D\}$. However, doing so leads to a deterministic network relying on undesirable sums of $\prod_{k=1}^L 2^{d_k}$ elements (see Appendix A.2 for details). Instead, we define a mapping $f_\theta \mapsto g_{\zeta(\theta)}$ which transforms the BAM network into a computation tree, as illustrated by Figure 4.

**BAM to tree architecture map.** Given a BAM network $f_\theta$ of $L$ layers with sizes $d_0, d_1, \ldots, d_L$ (reminder: $d_L{=}1$), we obtain a *computation tree* by decoupling the neurons (*i.e.*, the computation graph nodes): the tree leaves contain $\prod_{k=1}^L d_k$ copies of each of the $d_0$ BAM input neurons, and the tree root node corresponds to the single BAM output neuron. Each input-output path of the original BAM network becomes a path of length $L$ from one leaf to the tree root. Each tree edge has its own parameter (a real-valued scalar); the total number of edges is $D^\dagger := \sum_{k=0}^{L-1} d_k^\dagger$, with $d_k^\dagger := \prod_{i=k}^L d_i$. We define a set of tree parameters $\eta$ recursively according to the tree structure. From level $k$ to $k{+}1$, the tree has $d_k^\dagger$ edges. That is, each node at level $k{+}1$ has its own parameters subtree $\eta^{k+1} := \{\eta_i^k\}_{i=0}^{d_k}$, where each $\eta_i^k$ is either a weight vector containing the input edges parameters (by convention, $\eta_0^k \in \mathbb{R}^{d_{k-1}}$) or a parameter set (thus, $\eta_1^k, \ldots, \eta_{d_{k-1}}^k$ are themselves parameter subtrees).

Hence, the *deepest* elements of the recursive parameters set $\eta$ are weight vectors $\eta^1 \in \mathbb{R}^{d_0}$. Let us now define the output tree $g_\eta(\mathbf{x}) \coloneqq g_L(\mathbf{x}, \eta)$ on an input $\mathbf{x} \in \mathbb{R}^{d_0}$ as a recursive function:

$$g_1(\mathbf{x}, \{\mathbf{w}\}) = \text{sgn}\left(\mathbf{w} \cdot \mathbf{x}\right),$$

$$g_{k+1}(\mathbf{x}, \underbrace{\{\mathbf{w}, \eta_1^k, \dots, \eta_{d_k}^k\}}_{\eta^k}) = \text{sgn}\Big(\mathbf{w} \cdot \underbrace{(g_k(\mathbf{x}, \eta_1), \dots, g_k(\mathbf{x}, \eta_{d_k}))}_{\mathbf{g}_k(\mathbf{x}, \eta^k)}\Big) \text{ for } k = 1, \dots, L-1\,.$$

**BAM to tree parameters map.** Given BAM parameters $\theta$, we denote $\theta_{1:k} \coloneqq \text{vec}\big(\{\mathbf{W}_k\}_{i=1}^k\big)$. The mapping from $\theta$ into the corresponding (recursive) tree parameters set is $\zeta(\theta) = \{\mathbf{w}_L, \zeta_1(\theta_{1:L-1}), \dots, \zeta_{d_{L-1}}(\theta_{1:L-1})\}$, such that $\zeta_i(\theta_{1:k}) = \{\mathbf{w}_k^i, \zeta_1(\theta_{1:k-1}), \dots, \zeta_{d_{k-1}}(\theta_{1:k-1})\}$, and $\zeta_i(\theta_{1:1}) = \{\mathbf{w}_1^i\}$. Note that the parameters tree obtained by the transformation $\zeta(\theta)$ is highly redundant, as each weight vector $\mathbf{w}_k^i$ (the $i$th line of the $\mathbf{W}_k$ matrix from $\theta$) is replicated $d_{k+1}^\dagger$ times. This construction is such that $f_\theta(\mathbf{x}) = g_{\zeta(\theta)}(\mathbf{x})$ for all $\mathbf{x} \in \mathbb{R}^{d_0}$.

**Deterministic network.** With a slight abuse of notation, we let $\tilde{\eta} \sim Q_\eta \coloneqq \mathcal{N}(\eta, I_{D^\dagger})$ denote a parameter tree of the same structure as $\eta$, where every weight is sampled *iid* from a normal distribution. We denote $G_\theta(\mathbf{x}) \coloneqq \mathbb{E}_{\tilde{\eta} \sim Q_{\zeta(\theta)}} g_{\tilde{\eta}}(\mathbf{x})$, and we compute the output value of this predictor recursively. In the following, we denote $G_{\theta_{1:k+1}}^{(j)}(\mathbf{x})$ the function returning the $j$th neuron value of the layer $k+1$. Hence, the output of this network is $G_\theta(\mathbf{x}) = G_{\theta_{1:L}}^{(1)}(\mathbf{x})$. As such,

$$G_{\theta_{1:1}}^{(j)}(\mathbf{x}) = \int_{\mathbb{R}^{d_0}} Q_{\mathbf{w}_1^j}(\mathbf{v}) \text{sgn}(\mathbf{v} \cdot \mathbf{x}) d\mathbf{v} = \text{erf}\left(\frac{\mathbf{w}_1^j \cdot \mathbf{x}}{\sqrt{2}\|\mathbf{x}\|}\right),$$

$$G_{\theta_{1:k+1}}^{(j)}(\mathbf{x}) = \sum_{\mathbf{s} \in \{-1,1\}^{d_k}} \text{erf}\left(\frac{\mathbf{w}_{k+1}^j \cdot \mathbf{s}}{\sqrt{2 d_k}}\right) \Psi_{\mathbf{s}}^k(\mathbf{x}, \theta), \text{ with } \Psi_{\mathbf{s}}^k(\mathbf{x}, \theta) = \prod_{i=1}^{d_k} \left(\frac{1}{2} + \frac{1}{2} s_i \times G_{\theta_{1:k}}^{(i)}(\mathbf{x})\right). \quad (16)$$

The complete mathematical calculations leading to the above results are provided in Appendix A.3. The computation tree structure and the parameter mapping $\zeta(\theta)$ are crucial to obtain the recursive expression of Equation (16). However, note that this abstract mathematical structure is never manipulated explicitly. Instead, it allows computing each hidden layer vector $(G_{\theta_{1:k}}^{(j)}(\mathbf{x}))_{j=1}^{d_k}$ sequentially; a summation of $2^{d_k}$ terms is required for each layer $k = 1, \dots, L-1$.

**Stochastic approximation.** Following the Section 3.2 sampling procedure trick for the one hidden layer network, we propose to perform a stochastic approximation of the network prediction output, by a Monte Carlo sampling for each layer. Likewise, we recover exact and approximate derivatives in a layer-by-layer scheme. The related equations are given in Appendix A.4.

## 5  PBGNet: PAC-Bayesian SGD learning of binary activated networks

We design an algorithm to learn the parameters $\theta \in \mathbb{R}^D$ of the predictor $G_\theta$ by minimizing a PAC-Bayesian upper bound on the generalization loss $\mathcal{L}_\mathcal{D}(G_\theta)$. We name our algorithm PBGNet (**P**AC-Bayesian **B**inary **G**radient **Net**work), as it is a generalization of the PBGD (PAC-Bayesian Gradient Descent) learning algorithm for linear classifiers [Germain et al., 2009] to deep binary activated neural networks.

**Kullback-Leibler regularization.** The computation of a PAC-Bayesian bound value relies on two key elements: the empirical loss on the training set and the Kullback-Leibler divergence between the prior and the posterior. Sections 3 and 4 present exact computation and approximation schemes for the empirical loss $\widehat{\mathcal{L}}_S(G_\theta)$ (which is equal to $\widehat{\mathcal{L}}_S(F_\theta)$ when $L=2$). Equation (17) introduces the KL-divergence associated to the parameter maps of Section 4. We use the shortcut notation $\mathcal{K}(\theta, \mu)$ to refer to the divergence between two multivariate Gaussians of $D^\dagger$ dimensions, corresponding to learned parameters $\theta = \text{vec}\big(\{\mathbf{W}_k\}_{k=1}^L\big)$ and prior parameters $\mu = \text{vec}\big(\{\mathbf{U}_k\}_{k=1}^L\big)$.

$$\mathcal{K}(\theta, \mu) \coloneqq \text{KL}\Big(Q_{\zeta(\theta)} \,\big\|\, P_{\zeta(\mu)}\Big) = \frac{1}{2}\left(\|\mathbf{w}_L - \mathbf{u}_L\|^2 + \sum_{k=1}^{L-1} d_{k+1}^\dagger \big\|\mathbf{W}_k - \mathbf{U}_k\big\|_F^2\right), \quad (17)$$

where the factors $d_{k+1}^\dagger = \prod_{i=k+1}^L d_i$ are due to the redundancy introduced by transformation $\zeta(\cdot)$. This has the effect of penalizing more the weights on the first layers. It might have a considerable

influence on the bound value for very deep networks. On the other hand, we observe that this is consistent with the *fine-tuning* practice performed when training deep neural networks for a transfer learning task: prior parameters are learned on a first dataset, and the posterior weights are learned by adjusting the last layer weights on a second dataset [see Bengio, 2009, Yosinski et al., 2014].

**Bound minimization.** PBGNet minimizes the bound of Theorem 1 (rephrased as Equation 18). However, this is done indirectly by minimizing a variation on Theorem 2 and used in a deep learning context by Zhou et al. [2019] (Equation 19). Theorem 3 links both results (proof in Appendix A.5).

**Theorem 3.** *Given prior parameters $\mu \in \mathbb{R}^D$, with probability at least $1 - \delta$ over $S \sim \mathcal{D}^n$, we have for all $\theta$ on $\mathbb{R}^D$ :*

$$\mathcal{L}_{\mathcal{D}}(G_\theta) \leq \sup_{0 \leq p \leq 1} \left\{ p : \mathrm{kl}(\widehat{\mathcal{L}}_S(G_\theta) \| p) \leq \frac{1}{n}[\mathcal{K}(\theta, \mu) + \ln \frac{2\sqrt{n}}{\delta}] \right\} \tag{18}$$

$$= \inf_{C > 0} \left\{ \frac{1}{1 - e^{-C}} \left( 1 - \exp\left( -C \, \widehat{\mathcal{L}}_S(G_\theta) - \frac{1}{n}[\mathcal{K}(\theta, \mu) + \ln \frac{2\sqrt{n}}{\delta}] \right) \right) \right\}. \tag{19}$$

We use stochastic gradient descent (SGD) as the optimization procedure to minimize Equation (19) with respect to $\theta$ and $C$. It optimizes the same trade-off as in Equation (5), but choosing the $C$ value which minimizes the bound.[3] The originality of our SGD approach is that not only do we induce gradient randomness by selecting *mini-batches* among the training set $S$, we also approximate the loss gradient by sampling $T$ elements for the combinatorial sum at each layer. Our experiments show that, for some learning problems, reducing the sample size of the Monte Carlo approximation can be beneficial to the stochastic gradient descent. Thus the sample size value $T$ has an influence on the cost function space exploration during the training procedure (see Figure 7 in the appendix). Hence, we consider $T$ as a PBGNet hyperparameter.

## 6 Numerical experiments

Experiments were conducted on six binary classification datasets, described in Appendix B.

**Learning algorithms.** In order to get insights on the trade-offs promoted by the PAC-Bayes bound minimization, we compared PBGNet to variants focusing on empirical loss minimization. We train the models using multiple network architectures (depth and layer size) and hyperparameter choices. The objective is to evaluate the efficiency of our PAC-Bayesian framework both as a learning algorithm design tool and a model selection criterion. For all methods, the network parameters are trained using the Adam optimizer [Kingma and Ba, 2015]. Early stopping is used to interrupt the training when the cost function value is not improved for 20 consecutive epochs. Network architectures explored range from 1 to 3 hidden layers ($L$) and a hidden size $h \in \{10, 50, 100\}$ ($d_k = h$ for $1 \leq k < L$). Unless otherwise specified, the same randomly initialized parameters are used as a prior in the bound and as a starting point for SGD optimization [as in Dziugaite and Roy, 2017]. Also, for all models except MLP, we select the binary activation sampling size $T$ in a range going from 10 to 10000. More details about the experimental setting are given in Appendix B.

*MLP.* We compare to a standard network with $\tanh$ activation, as this activation resembles the $\mathrm{erf}$ function of PBGNet. We optimize the linear loss as the cost function and use 20% of training data as validation for hyperparameters selection. A weight decay parameter $\rho$ is selected between $0$ and $10^{-4}$. Using weight decay corresponds to adding an $L2$ regularizer $\frac{\rho}{2}\|\theta\|^2$ to the cost function, but contrary to the regularizer of Equation (17) promoted by PBGNet, this regularization is uniform for all layers.

*PBGNet$_\ell$.* This variant minimizes the empirical loss $\widehat{\mathcal{L}}(G_\theta)$, with an $L2$ regularization term $\frac{\rho}{2}\|\theta\|^2$. The corresponding weight decay $\rho$, as well as other hyperparameters, are selected using a validation set, exactly as the MLP does. The bound expression is not involved in the learning process and is computed on the model selected by the validation set technique.

*PBGNet$_{\ell\text{-bnd}}$.* Again, the empirical loss $\widehat{\mathcal{L}}(G_\theta)$ with an $L2$ regularization term $\frac{\rho}{2}\|\theta\|^2$ is minimized. However, only the weight decay hyperparameter $\rho$ is selected on the validation set, the other ones are selected by the bound. This method is motivated by an empirical observation: our PAC-Bayesian bound is a great model selection tool for most hyperparameters, except the weight decay term.

Table 1: Experiment results for the considered models on the binary classification datasets: error rates on the train and test sets ($E_S$ and $E_T$), and generalization bounds on the linear loss $\mathcal{L}_\mathcal{D}$ (Bnd). The PAC-Bayesian bounds hold with probability 0.95. Bound values for PBGNet$_\ell$ are trivial, excepted Adult with a bound value of 0.606, and are thus not reported. A visual representation of this table is presented in the appendix (Figure 5).

| Dataset | MLP | | PBGNet$_\ell$ | | PBGNet$_{\ell\text{-bnd}}$ | | | PBGNet | | | PBGNet$_{pre}$ | | |
|---|---|---|---|---|---|---|---|---|---|---|---|---|---|
| | $E_S$ | $E_T$ | $E_S$ | $E_T$ | $E_S$ | $E_T$ | Bnd | $E_S$ | $E_T$ | Bnd | $E_S$ | $E_T$ | Bnd |
| ads | 0.021 | 0.035 | 0.018 | **0.030** | 0.028 | 0.047 | 0.763 | 0.131 | 0.168 | 0.205 | 0.033 | 0.033 | 0.060 |
| adult | 0.137 | 0.152 | 0.133 | **0.149** | 0.147 | 0.155 | 0.281 | 0.154 | 0.163 | 0.214 | 0.149 | 0.154 | 0.164 |
| mnist17 | 0.002 | **0.004** | 0.003 | **0.004** | 0.004 | 0.006 | 0.096 | 0.005 | 0.007 | 0.040 | 0.004 | **0.004** | 0.010 |
| mnist49 | 0.004 | **0.013** | 0.003 | 0.018 | 0.029 | 0.035 | 0.311 | 0.035 | 0.040 | 0.139 | 0.016 | 0.017 | 0.028 |
| mnist56 | 0.004 | 0.013 | 0.003 | 0.011 | 0.022 | 0.024 | 0.172 | 0.022 | 0.025 | 0.090 | 0.009 | **0.009** | 0.018 |
| mnistLH | 0.006 | **0.018** | 0.004 | 0.019 | 0.046 | 0.051 | 0.311 | 0.049 | 0.052 | 0.160 | 0.026 | 0.027 | 0.033 |

*PBGNet.* As described in Section 5, the generalization bound is directly optimized as the cost function during the learning procedure and used solely for hyperparameters selection: no validation set is needed and all training data $S$ are exploited for learning.

*PBGNet$_{pre}$.* We also explore the possibility of using a part of the training data as a pre-training step. To do so, we split the training set into two halves. First, we minimize the empirical loss for a fixed number of 20 epochs on the first 50% of the training set. Then, we use the learned parameters as initialization and prior for PBGNet and learn on the second 50% of the training set.

**Analysis.** Results are summarized in Table 1, which highlights the strengths and weaknesses of the models. Both MLP and PBGNet$_\ell$ obtain competitive error scores but lack generalization guarantees. By introducing the bound value in the model selection process, even with the linear loss as the cost function, PBGNet$_{\ell\text{-bnd}}$ yields non-vacuous generalization bound values although with an increase in error scores. Using the bound expression for the cost function in PBGNet improves bound values while keeping similar performances. The Ads dataset is a remarkable exception where the small amount of training examples seems to radically constrain the network in the learning process as it hinders the KL divergence growth in the bound expression. With an informative prior from pre-training, PBGNet$_{pre}$ is able to recover competitive error scores while offering tight generalization guarantees. All selected hyperparameters are presented in the appendix (Table 4).

A notable observation is the impact of the bound exploitation for model selection on the train-test error gap. Indeed, PBGNet$_{\ell\text{-bnd}}$, PBGNet and PBGNet$_{pre}$ display test errors closer to their train errors, as compared to MLP and PBGNet$_\ell$. This behavior is more noticeable as the dataset size grows and suggests potential robustness to overfitting when the bound is involved in the learning process.

## 7   Conclusion and perspectives

We made theoretical and algorithmic contributions towards a better understanding of generalization abilities of binary activated multilayer networks, using PAC-Bayes. Note that the computational complexity of a learning epoch of PBGNet is higher than the cost induced in *binary neural networks* [Bengio, 2009, Hubara et al., 2016, 2017, Soudry et al., 2014]. Indeed, we focus on the optimization of the generalization guarantee more than computational complexity. Although we also propose a sampling scheme that considerably reduces the learning time required by our method, achieving a nontrivial tradeoff.

We intend to investigate how we could leverage the bound to learn suitable priors for PBGNet. Or equivalently, finding (from the bound point of view) the best network architecture. We also plan to extend our analysis to multiclass and multilabel prediction, and convolutional networks. We believe that this line of work is part of a necessary effort to give rise to a better understanding of the behavior of deep neural networks.

**Acknowledgments**

We would like to thank Mario Marchand for the insight leading to the Theorem 3, Gabriel Dubé and Jean-Samuel Leboeuf for their input on the theoretical aspects, Frédérik Paradis for his help with the implementation, and Robert Gower for his insightful comments. This work was supported in part by the French Project APRIORI ANR-18-CE23-0015, in part by NSERC and in part by Intact Financial Corporation. We gratefully acknowledge the support of NVIDIA Corporation with the donation of Titan Xp GPUs used for this research.

## Footnotes

[1]Bold uppercase letters denote matrices, bold lowercase letters denote vectors.

[2]We consider the activation function as an *element-wise* operator when applied to vectors or matrices.

[3]We also note that our training objective can be seen as a generalized Bayesian inference one [Knoblauch et al., 2019], where the tradeoff between the loss and the KL divergence is given by the PAC-Bayes bound.

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
