[Supplementary Material]

# A Supplementary Material

## A.1 From the *sign* activation to the *erf* function

For completeness, we present the detailed derivation of Equation (4). This result appears namely in Germain et al. [2009], Langford [2005], Langford and Shawe-Taylor [2002].

Given $\mathbf{x} \in \mathbb{R}^d$, we have

$$F_{\mathbf{w}}(\mathbf{x}) = \underset{\mathbf{v} \sim \mathcal{N}(\mathbf{w}, \mathrm{I})}{\mathbf{E}} \mathrm{sgn}(\mathbf{v} \cdot \mathbf{x})$$

$$= \int_{\mathbb{R}^d} \mathrm{sgn}(\mathbf{v} \cdot \mathbf{x}) \left( \frac{1}{\sqrt{2\pi}} \right)^d e^{-\frac{1}{2}\|\mathbf{v}-\mathbf{w}\|^2} d\mathbf{v}$$

$$= \int_{\mathbb{R}^d} \left( \mathbb{1}\left[\mathbf{v} \cdot \mathbf{x} > 0\right] - \mathbb{1}\left[\mathbf{v} \cdot \mathbf{x} < 0\right] \right) \left( \frac{1}{\sqrt{2\pi}} \right)^d e^{-\frac{1}{2}\|\mathbf{v}-\mathbf{w}\|^2} d\mathbf{v}$$

$$= \left( \frac{1}{\sqrt{2\pi}} \right)^d \int_{\mathbb{R}^d} \mathbb{1}\left[\mathbf{v} \cdot \mathbf{x} > 0\right] e^{-\frac{1}{2}\|\mathbf{v}-\mathbf{w}\|^2} d\mathbf{v} - \left( \frac{1}{\sqrt{2\pi}} \right)^d \int_{\mathbb{R}^d} \mathbb{1}\left[\mathbf{v} \cdot \mathbf{x} < 0\right] e^{-\frac{1}{2}\|\mathbf{v}-\mathbf{w}\|^2} d\mathbf{v} \,.$$

Without loss of generality, let us consider a vector basis where $\frac{\mathbf{x}}{\|\mathbf{x}\|}$ is the first coordinate. In this basis, the first elements of the vectors $\mathbf{v} = (v_1, v_2, \ldots, v_d)$ and $\mathbf{w} = (w_1, w_2, \ldots, w_d)$ are

$$v_1 = \frac{\mathbf{v} \cdot \mathbf{x}}{\|\mathbf{x}\|}, \qquad\qquad w_1 = \frac{\mathbf{w} \cdot \mathbf{x}}{\|\mathbf{x}\|} \,.$$

Hence, $\mathbf{v} \cdot \mathbf{x} = v_1 \cdot \|\mathbf{x}\|$ with $\|\mathbf{x}\| > 0$. Looking at the left side of the subtraction from the previous equation, we thus have

$$\left( \frac{1}{\sqrt{2\pi}} \right)^d \int_{\mathbb{R}^d} \mathbb{1}\left[\mathbf{v} \cdot \mathbf{x} > 0\right] e^{-\frac{1}{2}\|\mathbf{v}-\mathbf{w}\|^2} d\mathbf{v}$$

$$= \int_{\mathbb{R}} \mathbb{1}\left[v_1 > 0\right] \frac{1}{\sqrt{2\pi}} e^{-\frac{1}{2}(v_1-w_1)^2} \left[ \int_{\mathbb{R}^{d-1}} \left( \frac{1}{\sqrt{2\pi}} \right)^{d-1} e^{-\frac{1}{2}\|\mathbf{v}_{2:d}-\mathbf{w}_{2:d}\|^2} d\mathbf{v}_{2:d} \right] dv_1$$

$$= \frac{1}{\sqrt{2\pi}} \int_{-\infty}^{\infty} \mathbb{1}\left[t > -w_1\right] e^{-\frac{1}{2}t^2} dt \,,$$

with $t := v_1 - w_1$. Hence,

$$\underset{\mathbf{v} \sim \mathcal{N}(\mathbf{w}, \mathrm{I})}{\mathbf{E}} \mathrm{sgn}(\mathbf{v} \cdot \mathbf{x}) = \frac{1}{\sqrt{2\pi}} \int_{-\infty}^{\infty} \mathbb{1}\left[t > -w_1\right] e^{-\frac{1}{2}t^2} dt - \frac{1}{\sqrt{2\pi}} \int_{-\infty}^{\infty} \mathbb{1}\left[t < -w_1\right] e^{-\frac{1}{2}t^2} dt$$

$$= \frac{1}{\sqrt{2\pi}} \int_{-w_1}^{\infty} e^{-\frac{1}{2}t^2} dt - \frac{1}{\sqrt{2\pi}} \int_{-\infty}^{-w_1} e^{-\frac{1}{2}t^2} dt$$

$$= \frac{1}{2} + \frac{1}{\sqrt{2\pi}} \int_{0}^{w_1} e^{-\frac{1}{2}t^2} dt - \frac{1}{2} + \frac{1}{\sqrt{2\pi}} \int_{0}^{w_1} e^{-\frac{1}{2}t^2} dt$$

$$= \frac{\sqrt{2}}{\sqrt{\pi}} \int_{0}^{w_1} e^{-\frac{1}{2}t^2} dt$$

$$= \frac{2}{\sqrt{\pi}} \int_{0}^{\frac{w_1}{\sqrt{2}}} e^{-u^2} du \quad \text{with } u = \frac{t}{\sqrt{2}}$$

$$= \mathrm{erf}\left( \frac{w_1}{\sqrt{2}} \right)$$

$$= \mathrm{erf}\left( \frac{\mathbf{w} \cdot \mathbf{x}}{\sqrt{2}\,\|\mathbf{x}\|} \right) \,,$$

where $\mathrm{erf}(\cdot)$ is the Gauss error function defined as $\mathrm{erf}(x) = \frac{2}{\sqrt{\pi}} \int_{0}^{x} e^{-t^2} dt$.

Figure 3: Visual comparison between the erf, tanh and sgn activation functions.

## A.2 Aggregation of multilayer networks without the tree architecture

To understand the benefit of the **BAM to tree architecture map** introduced in Section 4, let us compute the prediction function of an aggregation of two hidden layers networks following the same strategy as for the single hidden layer case (see Subsection 3.1).

The two hidden layer network is parameterized by weights $\theta = \text{vec}(\{\mathbf{W}_1, \mathbf{W}_2, \mathbf{w}_3\})$, with $\mathbf{W}_1 \in \mathbb{R}^{d_1 \times d_0}$, $\mathbf{W}_2 \in \mathbb{R}^{d_2 \times d_1}$ and $\mathbf{w}_3 \in \mathbb{R}^{d_2}$. Given an input $\mathbf{x} \in \mathbb{R}^{d_0}$, the output of the network is

$$f_\theta(\mathbf{x}) = \text{sgn}\big(\mathbf{w}_3 \cdot \text{sgn}(\mathbf{W}_2 \cdot \text{sgn}(\mathbf{W}_1 \mathbf{x}))\big). \tag{20}$$

We seek to compute $F_\theta(\mathbf{x}) = \mathbf{E}_{\tilde{\theta} \sim Q_\theta} f_{\tilde{\theta}}(\mathbf{x})$ with $\tilde{\theta} = \text{vec}(\{\mathbf{V}_1, \mathbf{V}_2, \mathbf{v}_3\})$ and $Q_\theta = \mathcal{N}(\theta, I_D)$. First, we need to decompose the probability of each $\tilde{\theta} \sim Q_\theta$ as $Q_\theta(\tilde{\theta}) = Q_1(\mathbf{V}_1) Q_2(\mathbf{V}_2) Q_3(\mathbf{v}_3)$, with $Q_1 = \mathcal{N}(\mathbf{W}_1, I_{d_0 d_1})$, $Q_2 = \mathcal{N}(\mathbf{W}_2, I_{d_1 d_2})$ and $Q_3 = \mathcal{N}(\mathbf{w}_3, I_{d_2})$.

$$F_\theta(\mathbf{x}) = \mathop{\mathbf{E}}_{\tilde{\theta} \sim Q_\theta} f_{\tilde{\theta}}(\mathbf{x})$$

$$= \int_{\mathbb{R}^{d_1 \times d_0}} Q_1(\mathbf{V}_1) \int_{\mathbb{R}^{d_2 \times d_1}} Q_2(\mathbf{V}_2) \int_{\mathbb{R}^{d_2}} Q_3(\mathbf{v}_3) \text{sgn}(\mathbf{v}_3 \cdot \text{sgn}(\mathbf{V}_2 \text{sgn}(\mathbf{V}_1 \mathbf{x}))) d\mathbf{v}_3 d\mathbf{V}_2 d\mathbf{V}_1$$

$$= \int_{\mathbb{R}^{d_1 \times d_0}} Q_1(\mathbf{V}_1) \int_{\mathbb{R}^{d_2 \times d_1}} Q_2(\mathbf{V}_2) \, \text{erf}\left(\frac{\mathbf{w}_3 \cdot \text{sgn}(\mathbf{V}_2 \text{sgn}(\mathbf{V}_1 \mathbf{x}))}{\sqrt{2}\|\text{sgn}(\mathbf{V}_2 \text{sgn}(\mathbf{V}_1 \mathbf{x}))\|}\right) d\mathbf{V}_2 d\mathbf{V}_1$$

$$= \sum_{\mathbf{t} \in \{-1,1\}^{d_2}} \text{erf}\left(\frac{\mathbf{w}_3 \cdot \mathbf{t}}{\sqrt{2d_2}}\right) \int_{\mathbb{R}^{d_1 \times d_0}} Q_1(\mathbf{V}_1) \int_{\mathbb{R}^{d_2 \times d_1}} Q_2(\mathbf{V}_2) \, \mathbb{1}[\mathbf{t} = \text{sgn}(\mathbf{V}_2 \text{sgn}(\mathbf{V}_1 \mathbf{x}))] \, d\mathbf{V}_2 d\mathbf{V}_1$$

$$= \sum_{\mathbf{t} \in \{-1,1\}^{d_2}} \text{erf}\left(\frac{\mathbf{w}_3 \cdot \mathbf{t}}{\sqrt{2d_2}}\right) \int_{\mathbb{R}^{d_1 \times d_0}} Q_1(\mathbf{V}_1) \prod_{i=1}^{d_2} \int_{\mathbb{R}^{d_2 \times d_1}} Q_2^i(\mathbf{v}_2^i) \, \mathbb{1}[t_i \text{sgn}(\mathbf{v}_2^i \text{sgn}(\mathbf{V}_1 \mathbf{x})) > 0] \, d\mathbf{v}_2^i d\mathbf{V}_1$$

$$= \sum_{\mathbf{t} \in \{-1,1\}^{d_2}} \text{erf}\left(\frac{\mathbf{w}_3 \cdot \mathbf{t}}{\sqrt{2d_2}}\right) \sum_{\mathbf{s} \in \{-1,1\}^{d_1}} \int_{\mathbb{R}^{d_1 \times d_0}} Q_1(\mathbf{V}_1) \mathbb{1}[\mathbf{s} = \text{sgn}(\mathbf{V}_1 \mathbf{x})] d\mathbf{V}_1 \prod_{i=1}^{d_2} \left[\frac{1}{2} + \frac{t_i}{2} \text{erf}\left(\frac{\mathbf{w}_2^i \cdot \mathbf{s}}{\sqrt{2d_1}}\right)\right]$$

$$= \sum_{\mathbf{t} \in \{-1,1\}^{d_2}} \text{erf}\left(\frac{\mathbf{w}_3 \cdot \mathbf{t}}{\sqrt{2d_2}}\right) \sum_{\mathbf{s} \in \{-1,1\}^{d_1}} \underbrace{\prod_{j=1}^{d_1} \left[\frac{1}{2} + \frac{s_j}{2} \text{erf}\left(\frac{\mathbf{w}_1^j \cdot \mathbf{x}}{\sqrt{2}\|\mathbf{x}\|}\right)\right]}_{\Psi_\mathbf{s}(\mathbf{x}, \mathbf{W}_1)} \underbrace{\prod_{i=1}^{d_2} \left[\frac{1}{2} + \frac{t_i}{2} \text{erf}\left(\frac{\mathbf{w}_2^i \cdot \mathbf{s}}{\sqrt{2d_1}}\right)\right]}_{\Psi_\mathbf{t}(\mathbf{s}, \mathbf{W}_2)}$$

For each combination of first layer activation values $\mathbf{s} \in \{-1, 1\}^{d_1}$ and second layer activation values $\mathbf{t} \in \{-1, 1\}^{d_2}$, one needs to compute its probability $\Psi_\mathbf{s}(\mathbf{x}, \mathbf{W}_1)\Psi_\mathbf{t}(\mathbf{s}, \mathbf{W}_2)$. This leads to a summation of $2^{d_1} \times 2^{d_2}$ terms. Instead, the layer by layer computation obtained by our tree mapping trick implies $2^{d_1} + 2^{d_2}$ terms. This strategy enables a forward computation similar to traditional neural network, as each hidden layer values relies solely on the values of the previous layer.

## A.3 Prediction for the multilayer case (with the proposed tree architecture map)

Details of the complete mathematical calculations leading to Equation (16) are presented below:

$$
\begin{aligned}
G^{(j)}_{\theta_{1:k+1}}(\mathbf{x}) &:= \int Q_{\zeta_j(\theta_{1:k+1})}(\tilde{\eta})\, g_{k+1}(\mathbf{x},\tilde{\eta})\, d\tilde{\eta} \\
&= \int Q_{\zeta(\theta_{1:k})}(\tilde{\eta}_1)\dots \int Q_{\zeta(\theta_{1:k})}(\tilde{\eta}_{d_k}) \left( \int_{\mathbb{R}^{d^k}} Q_{\mathbf{w}^j_k}(\mathbf{v})\mathrm{sgn}[\mathbf{v}\cdot \mathbf{g}_k(\mathbf{x},\tilde{\eta})]d\mathbf{v} \right) d\tilde{\eta}_{d_k}\dots d\tilde{\eta}_1 \\
&= \int Q_{\zeta(\theta_{1:k})}(\tilde{\eta}_1)\dots \int Q_{\zeta(\theta_{1:k})}(\tilde{\eta}_{d_k}) \,\mathrm{erf}\left( \frac{\mathbf{w}^j_{k+1}\cdot \mathbf{g}_k(\mathbf{x},\tilde{\eta})}{\sqrt{2}\|\mathbf{g}_k(\mathbf{x},\tilde{\eta})\|} \right) d\tilde{\eta}_{d_k}\dots d\tilde{\eta}_1 \\
&= \sum_{\mathbf{s}\in\{-1,1\}^{d_k}} \mathrm{erf}\left( \frac{\mathbf{w}^j_{k+1}\cdot \mathbf{s}}{\sqrt{2d_k}} \right) \int Q_{\zeta(\theta_{1:k})}(\tilde{\eta}_1)\dots \int Q_{\zeta(\theta_{1:k})}(\tilde{\eta}_{d_k}) \mathbb{1}[\mathbf{s}=\mathbf{g}_k(\mathbf{x},\tilde{\eta})]d\tilde{\eta}_{d_k}\cdots d\tilde{\eta}_1 \\
&= \sum_{\mathbf{s}\in\{-1,1\}^{d_k}} \mathrm{erf}\left( \frac{\mathbf{w}^j_{k+1}\cdot \mathbf{s}}{\sqrt{2d_k}} \right) \prod_{i=1}^{d_k} \int Q_{\zeta(\theta_{1:k})}(\tilde{\eta}_i) \mathbb{1}[s_i = g_k(\mathbf{x},\tilde{\eta}_i)]d\tilde{\eta}_i \\
&= \sum_{\mathbf{s}\in\{-1,1\}^{d_k}} \mathrm{erf}\left( \frac{\mathbf{w}^j_{k+1}\cdot \mathbf{s}}{\sqrt{2d_k}} \right) \prod_{i=1}^{d_k} \int Q_{\zeta(\theta_{1:k})}(\tilde{\eta}_i) \left( \frac{1}{2} + \frac{s_i}{2} g_k(\mathbf{x},\tilde{\eta}_i) \right) d\tilde{\eta}_i \\
&= \sum_{\mathbf{s}\in\{-1,1\}^{d_k}} \mathrm{erf}\left( \frac{\mathbf{w}^j_{k+1}\cdot \mathbf{s}}{\sqrt{2d_k}} \right) \prod_{i=1}^{d_k} \left( \frac{1}{2} + \frac{s_i}{2} \int Q_{\zeta(\theta_{1:k})}(\tilde{\eta}_i) g_k(\mathbf{x},\tilde{\eta}_i)d\tilde{\eta}_i \right) \\
&= \sum_{\mathbf{s}\in\{-1,1\}^{d_k}} \mathrm{erf}\left( \frac{\mathbf{w}^j_{k+1}\cdot \mathbf{s}}{\sqrt{2d_k}} \right) \underbrace{\prod_{i=1}^{d_k} \left( \frac{1}{2} + \frac{1}{2} s_i \times G^{(i)}_{\theta_{1:k}}(\mathbf{x}) \right)}_{\Psi^k_{\mathbf{s}}(\mathbf{x},\eta)}.
\end{aligned}
$$

Moreover,

$$
\Psi^k_{\mathbf{s}}(\mathbf{x},\theta) = \prod_{i=1}^{d_k} \underbrace{\left( \frac{1}{2} + \frac{1}{2} s_i \times G^{(i)}_{\theta_{1:k}}(\mathbf{x}) \right)}_{\psi^k_{s_i}(\mathbf{x},\theta)}.
$$

Base case:

$$
\begin{aligned}
G^{(j)}_{\theta_{1:1}}(\mathbf{x}) &= \mathop{\mathbb{E}}_{\tilde{\eta}\sim\mathcal{N}(\zeta_j(\theta_{1:1}),I)} g_1(\mathbf{x},\tilde{\eta}) \\
&= \int_{\mathbb{R}^{d_0}} Q_{\mathbf{w}^j_1}(\mathbf{v})\mathrm{sgn}(\mathbf{v}\cdot \mathbf{x})d\mathbf{v} \\
&= \mathrm{erf}\left( \frac{\mathbf{w}^j_1\cdot \mathbf{x}}{\sqrt{2}\|\mathbf{x}\|} \right).
\end{aligned}
$$

## A.4 Derivatives of the multilayer case (with the proposed tree architecture map)

We first aim at computing $\frac{\partial}{\partial \mathbf{w}_{k+1}} G_{\theta_{1:k+1}}(\mathbf{x})$.

Recall that $\mathbf{w}_k^j \in \{\mathbf{w}_k^1, \ldots, \mathbf{w}_k^{d_k}\}$ is the $j^{\text{th}}$ line of $\mathbf{W}_k$, that is the input weights of the corresponding hidden layer's neuron.

$$\frac{\partial}{\partial \mathbf{w}_{k+1}^j} G_{\theta_{1:k+1}}^{(j)}(\mathbf{x}) = \frac{\partial}{\partial \mathbf{w}_{k+1}^j} \sum_{\mathbf{s} \in \{-1,1\}^{d_k}} \operatorname{erf}\left(\frac{\mathbf{w}_{k+1}^j \cdot \mathbf{s}}{\sqrt{2d_k}}\right) \Psi_{\mathbf{s}}^k(\mathbf{x}, \theta)$$

$$= \sum_{\mathbf{s} \in \{-1,1\}^{d_k}} \frac{\mathbf{s}}{\sqrt{2d_k}} \operatorname{erf}'\left(\frac{\mathbf{w}_{k+1}^j \cdot \mathbf{s}}{\sqrt{2d_k}}\right) \Psi_{\mathbf{s}}^k(\mathbf{x}, \theta).$$

The base case of the recursion is

$$\frac{\partial}{\partial \mathbf{w}_1^j} G_{\theta_{1:1}}^{(j)}(\mathbf{x}) = \frac{\partial}{\partial \mathbf{w}_1^j} \operatorname{erf}\left(\frac{\mathbf{w}_1^j \cdot \mathbf{x}}{\sqrt{2}\|\mathbf{x}\|}\right)$$

$$= \frac{\mathbf{x}}{\sqrt{2}\|\mathbf{x}\|} \operatorname{erf}'\left(\frac{\mathbf{w}_1^j \cdot \mathbf{x}}{\sqrt{2}\|\mathbf{x}\|}\right).$$

In order to propagate the error through the layers, we also need to compute for $k > 1$:

$$\frac{\partial}{\partial G_{\theta_{1:k}}^{(l)}} G_{\theta_{1:k+1}}^{(j)} = \frac{\partial}{\partial G_{\theta_{1:k}}^{(l)}} \sum_{\mathbf{s} \in \{-1,1\}^{d_k}} \operatorname{erf}\left(\frac{\mathbf{w}_{k+1}^j \cdot \mathbf{s}}{\sqrt{2d_k}}\right) \prod_{i=1}^{d_k} \left(\frac{1}{2} + \frac{1}{2} s_i \times G_{\theta_{1:k}}^{(i)}\right)$$

$$= \sum_{\mathbf{s} \in \{-1,1\}^{d_k}} \operatorname{erf}\left(\frac{\mathbf{w}_{k+1}^j \cdot \mathbf{s}}{\sqrt{2d_k}}\right) \left[\frac{\Psi_{\mathbf{s}}^k(\mathbf{x}, \theta)}{\psi_{s_l}^k(\mathbf{x}, \theta)}\right] \frac{\partial}{\partial G_{\theta_{1:k}}^{(l)}} \psi_{s_l}^k(\mathbf{x}, \theta)$$

$$= \sum_{\mathbf{s} \in \{-1,1\}^{d_k}} \operatorname{erf}\left(\frac{\mathbf{w}_{k+1}^j \cdot \mathbf{s}}{\sqrt{2d_k}}\right) \left[\frac{s_l \Psi_{\mathbf{s}}^k(\mathbf{x}, \theta)}{2\psi_{s_l}^k(\mathbf{x}, \theta)}\right].$$

Thus, we can compute

$$\frac{\partial \widehat{\mathcal{L}}_S\left(G_{\theta_{1:k+1}}^{(j)}(\mathbf{x})\right)}{\partial \mathbf{w}_k^j} = \sum_l \frac{\partial \widehat{\mathcal{L}}_S\left(G_{\theta_{1:k+1}}^{(j)}(\mathbf{x})\right)}{\partial G_{\theta_{1:k+1}}^{(l)}} \frac{\partial G_{\theta_{1:k+1}}^{(l)}}{\partial G_{\theta_{1:k}}^{(j)}} \frac{\partial G_{\theta_{1:k}}^{(j)}}{\partial \mathbf{w}_k^j}.$$

## A.5 Proof of Theorem 3

**Lemma 4** (Germain et al. [2009], Proposition 2.1; Lacasse [2010], Proposition 6.2.2).
*For any $0 \leq q \leq p < 1$, we have*

$$\sup_{C>0} \left[ \Delta(C, q, p) \right] = \mathrm{kl}(q\|p) \,,$$

*with*

$$\Delta(C, q, p) := -\ln(1 - p(1 - e^{-C})) - Cq \,. \tag{21}$$

*Proof.* For $0 \leq q, p < 1$, $\Delta(C, q, p)$ is concave in $C$ and the maximum is $c_0 = -\ln\left(\frac{qp-p}{qp-q}\right)$. Moreover, $\Delta(c_0, q, p) = \mathrm{kl}(q\|p)$. $\qquad\square$

**Theorem 3.** *Given prior parameters $\mu \in \mathbb{R}^D$, with probability at least $1 - \delta$ over $S \sim \mathcal{D}^n$, we have for all $\theta$ on $\mathbb{R}^D$ :*

$$\mathcal{L}_{\mathcal{D}}(G_\theta) \leq \sup_{0 \leq p \leq 1} \left\{ p : \mathrm{kl}(\widehat{\mathcal{L}}_S(G_\theta)\|p) \leq \frac{1}{n}[\mathcal{K}(\theta, \mu) + \ln\tfrac{2\sqrt{n}}{\delta}] \right\}$$

$$= \inf_{C>0} \left\{ \frac{1}{1 - e^{-C}} \left( 1 - \exp\left( -C\,\widehat{\mathcal{L}}_S(G_\theta) - \frac{1}{n}[\mathcal{K}(\theta, \mu) + \ln\tfrac{2\sqrt{n}}{\delta}] \right) \right) \right\} \,.$$

*Proof.* In the following, we denote $\xi := \frac{1}{n}[\mathcal{K}(\theta, \mu) + \ln\frac{2\sqrt{n}}{\delta}]$ and we assume $0 < \xi < \infty$. Let us define

$$p^* := \sup_{0 \leq p \leq 1} \left\{ p : \mathrm{kl}(\widehat{\mathcal{L}}_S(G_\theta)\|p) \leq \xi \right\}. \tag{22}$$

First, by a straightforward rewriting of Theorem 1 [Seeger, 2002], we have, with probability at least $1 - \delta$ over $S \sim \mathcal{D}^n$,

$$\mathcal{L}_{\mathcal{D}}(G_\theta) \leq p^* \,.$$

Then, we want to show

$$p^* = \inf_{C>0} \left\{ \frac{1}{1 - e^{-C}} \left( 1 - \exp\left( -C\,\widehat{\mathcal{L}}_S(G_\theta) - \xi \right) \right) \right\}. \tag{23}$$

**Case $\widehat{\mathcal{L}}_S(G_\theta) < 1$:** The function $\mathrm{kl}(\widehat{\mathcal{L}}_S(G_\theta)\|p)$ is strictly increasing for $p > \widehat{\mathcal{L}}_S(G_\theta)$. Thus, the supremum value $p^*$ is always reached in Equation (22), so we have

$$\mathrm{kl}(\widehat{\mathcal{L}}_S(G_\theta)\|p^*) = \xi,$$

and, by Lemma 4,

$$\sup_{C>0} \left[ \Delta(C, \widehat{\mathcal{L}}_S(G_\theta), p^*) \right] = \xi \,, \tag{24}$$

$$\text{and } \forall C > 0 : \Delta(C, \widehat{\mathcal{L}}_S(G_\theta), p^*) \leq \xi \,. \tag{25}$$

Let $C^* := \mathrm{argsup}_{C>0} \left[ \Delta(C, \widehat{\mathcal{L}}_S(G_\theta), p^*) \right]$.

By rearranging the terms of $\Delta(C^*, \widehat{\mathcal{L}}_S(G_\theta), p^*)$ (see Equation 21), we obtain, from Line (24),

$$p^* = \frac{1}{1 - e^{-C^*}} \left( 1 - \exp\left( -C^*\,\widehat{\mathcal{L}}_S(G_\theta) - \xi \right) \right), \tag{26}$$

and, from Line (25),

$$\forall C > 0 : p^* \leq \frac{1}{1 - e^{-C}} \left( 1 - \exp\left( -C\,\widehat{\mathcal{L}}_S(G_\theta) - \xi \right) \right). \tag{27}$$

Thus, combining Lines (26) and (27), we finally prove the desired result of Equation (23).

**Case $\widehat{\mathcal{L}}_S(G_\theta) = 1$:** From Equation (22), we have $p^* = 1$, because $\lim_{p \to 1} \mathrm{kl}(1\|p) = 0$. We also have $\frac{1 - e^{-C-\xi}}{1 - e^{-C}} \geq 1$ and $\lim_{C \to \infty} \left[ \frac{1 - e^{-C-\xi}}{1 - e^{-C}} \right] = 1$, thus fulfilling Equation (23). $\qquad\square$

Table 2: Datasets overview.

| Dataset | $n_{\text{train}}$ | $n_{\text{test}}$ | $d$ |
|---|---|---|---|
| ads | 2459 | 820 | 1554 |
| adult | 36631 | 12211 | 108 |
| mnist17 | 11377 | 3793 | 784 |
| mnist49 | 10336 | 3446 | 784 |
| mnist56 | 9891 | 3298 | 784 |
| mnistLH | 52500 | 17500 | 784 |

Table 3: Models overview.

| Model name | Cost function | Train split | Valid split | Model selection | Prior |
|---|---|---|---|---|---|
| MLP | linear loss, L2 regularized | 80% | 20% | valid linear loss | - |
| PBGNet$_\ell$ | linear loss, L2 regularized | 80% | 20% | valid linear loss | random init |
| PBGNet$_{\ell\text{-bnd}}$ | linear loss, L2 regularized | 80% | 20% | hybrid (see B.2) | random init |
| PBGNet | PAC-Bayes bound | 100 % | - | PAC-Bayes bound | random init |
| PBGNet$_{\text{pre}}$ | | | | | |
| – pretrain | linear loss (20 epochs) | 50% | - | - | random init |
| – final | PAC-Bayes bound | 50% | - | PAC-Bayes bound | pretrain |

# B  Experiments

## B.1  Datasets

In Section 6 we use the datasets Ads (a small dataset related to advertisements on web pages), Adult (a low-dimensional task about predicting income from census data) and four binary variants of the MNIST handwritten digits:

**ads**  http://archive.ics.uci.edu/ml/datasets/Internet+Advertisements
    The first 4 features which have missing values are removed.

**adult**  https://archive.ics.uci.edu/ml/datasets/Adult

**mnist**  http://yann.lecun.com/exdb/mnist/
    Binary classification tasks are compiled with the following pairs:

- Digits pairs 1 vs. 7, 4 vs. 9 and 5 vs. 6.
- Low digits vs. high digits ($\{0, 1, 2, 3, 4\}$ vs $\{5, 6, 7, 8, 9\}$) identified as *mnistLH*.

We split the datasets into training and testing sets with a 75/25 ratio. Table 2 presents an overview of the datasets statistics.

## B.2  Learning algorithms details

Table 3 summarizes the characteristics of the learning algorithms used in the experiments.

**Cost functions.**
MLP, PBGNet$_\ell$, PBGNet$_{\ell\text{-bnd}}$ : The gradient descent minimizes the following according to parameters $\theta$.

$$\widehat{\mathcal{L}}_S(G_\theta) + \frac{\rho}{2}\|\theta\|^2 \,,$$

where $\rho$ is the L2 regularization "weight decay" penalization term.

PBGNet, PBGNet$_{\text{pre}}$ : The gradient descent minimizes the following according to parameters $\theta$ and $C \in \mathbb{R}^+$.

$$\frac{1}{1-e^{-C}} \left( 1 - \exp\left( -C\,\widehat{\mathcal{L}}_S(G_\theta) - \frac{1}{n}\left[\mathcal{K}(\theta,\mu) + \ln\frac{2\sqrt{n}}{\delta}\right]\right)\right) \,,$$

where $\mathcal{K}(\theta,\mu)$ is given by Equation (17).

**Hyperparameter choices.**    We execute each learning algorithm for combination of hyperparameters selected among the following values.

- Hidden layers $\in \{1, 2, 3\}$.
- Hidden size $\in \{10, 50, 100\}$.
- Sample size $\in \{10, 50, 100, 1000, 10000\}$.
- Weight decay $\in \{0, 10^{-4}, 10^{-6}\}$.
- Learning rate $\in \{0.1, 0.01, 0.001\}$.

Note that the sample size does not apply to MLP and weight decay is set to 0 for PBGNet and PBGNet$_{pre}$). For the learning algorithms that use the PAC-Bayes bound for model selection, the union bound is applied to compute a valid bound value considering the 9 possible combinations of "Hidden size" and "Hidden layers" hyperparameters: we set $\delta = \frac{0.05}{9}$ in Theorem 3 such that the selected model bound value holds with probability 0.95.

We report all the hyperparameters of selected models, for all learning algorithms and all datasets, in Table 4. The errors and bounds for these selected models are presented by Table 1 of the main paper. The same results are visually illustrated by Figure 5.

**Hybrid model selection scheme.**    Of note, PBGNet$_{\ell\text{-bnd}}$ has a unique hyperparameters selection approach using a combination of the validation loss and the bound value. First all hyperparameters, except the weight decay, are selected in order to minimize the bound value. This includes choosing the best epoch from which loading the network weights. Thus, we obtain the best models according to the bound for each weight decay values considered. Then, the validation loss can be used to identify the best model between those, hence selecting the weight decay value.

**Optimization.**    For all methods, the network parameters are trained using the Adam optimizer [Kingma and Ba, 2015] for a maximum of 150 epochs on mini-batches of size 32 for the smaller datasets (Ads and MNIST digit pairs) and size 64 for bigger datasets (Adult and mnistLH). Initial learning rate is selected in $\{0.1, 0.01, 0.001\}$ and halved after each 5 consecutive epochs without a decrease in the cost function value. We empirically observe that the prediction accuracy of PBGNet is usually better when trained using Adam optimizer than with *plain* stochastic gradient descent, while both optimization algorithms give comparable results for our MLP model. The study of this phenomenon is considered as an interesting future research direction.

Usually in deep learning framework training loops, the empirical loss of an epoch is computed as the averaged loss of each mini-batch. As the weights are updated after each mini-batch, the resulting epoch loss is only an approximation for the empirical loss of the final mini-batch weights. The linear loss being a significant element of the PAC-Bayesian bound expression, the approximation has a non-negligible impact over the corresponding bound value. One could obtain the accurate empirical loss for each epoch by assessing the network performance on the complete training data at the end of each epoch. We empirically evaluated that doing so leads to an increase of about a third of the computational cost per epoch for the inference computation. A practical alternative used in our experiments is to simply rely on the averaged empirical loss on the mini-batches in the bound expression for epoch-related actions: learning rate reduction, early stopping and best epoch selection.

**Prediction.**    Once the best epoch is selected, we can afford to compute the correct empirical loss for those weights and use it to obtain the corresponding bound value. However, because PBGNet and its variants use a Monte Carlo approximation in the inference stage, the predicted output is not deterministic. Thus, to obtain the values reported in Table 1, we repeated the prediction over the training data 20 times for the empirical loss computation of the selected epoch. The inference repetition process was also carried out on the testing set, hence reported values $E_S$, $E_T$ and Bnd of the results consist in the mean over 20 approximated predictions. The standard deviations are consistently below $0.001$, with the exception of PBGNet$_{\ell\text{-bnd}}$ on Ads for $E_T$ which has a standard deviation of $0.00165$. If network prediction consistency is crucial, one can set a higher sample size during inference to decrease variability, but keep a smaller sample size during training to reduce computational costs.

Figure 4: Illustration of the proposed method in Section 3 for a one hidden layer network of size $d_1 = 3$, interpreted as a majority vote over 8 binary representations $\mathbf{s} \in \{-1, 1\}^3$. For each $\mathbf{s}$, a plot shows the values of $F_{\mathbf{w}_2}(\mathbf{s})\Psi_{\mathbf{s}}(\mathbf{x}, \mathbf{W}_1)$. The sum of these values gives the deterministic network output $F_\theta(\mathbf{x})$ (see Equation 9). We also show the BAM network output $f_\theta(\mathbf{x})$ for the same parameters $\theta$ (see Equation 6).

**Implementation details.** We implemented PBGNet using `PyTorch` library [Paszke et al., 2017], using the `Poutyne` framework [Paradis, 2018] for managing the networks training workflow. The code used to run the experiments is available at:

https://github.com/gletarte/dichotomize-and-generalize

When computing the full combinatorial sum, a straightforward implementation is feasible, gradients being computed efficiently by the automatic differentiation mechanism. For speed purposes, we compute the sum as a matrix operation by loading all $\mathbf{s} \in \{-1, 1\}^{d_k}$ as an array. Thus we are mainly limited by memory usage on the GPU, a single hidden layer of hidden size 20 using approximately 10Gb of memory depending on the dataset and batch size used.

For the Monte Carlo approximation, we need to insert the gradient approximation in a flexible way into the derivative graph of the automatic differentiation mechanism. Therefore, we implemented each layer as a function of the weights and the output of the previous layer, with explicit forward and backward expression[4]. Thus the automatic differentiation mechanism is able to accurately propagate the gradient through our approximated layers, and also combine gradient from other sources towards the weights (for example the gradient from the KL computation when optimizing with the bound as the cost function).

Experiments were performed on NVIDIA GPUs (Tesla V100, Titan Xp, GeForce GTX 1080 Ti).

## B.3 Additional results

Figure 4 reproduces the experiment presented by Figure 1 with another toy dataset. Table 5 exhibits a variance analysis of Table 1. Figure 6 shows the impact of the training set size. Figure 7 studies the effect of the sampling size $T$ on the stochastic gradient descent procedure. See figure/table captions for details.

Figure 5: Visualization of experiment results for the models on the binary classification datasets. The colored bars display the test error while the black outlined bars exhibit the train error. The PAC-Bayesian bounds are identified on the top of the bars and hold with probability 0.95.

Figure 6: Study of the training sample size effect on PBGNet$_\ell$ and PBGNet for the biggest dataset, mnistLH. The middle column report results for PBGNet$_\ell$ with a fixed network architecture of a single hidden layer of 10 neurons, for a direct comparison with PBGNet which always selects this architecture. For each training set size values, 10 repetitions of the learning procedure with different random seeds were performed: each of them executed on different (random) train/test/valid dataset splits, and the stochastic gradient descent is initialized with different random weights. Metrics means of the learned models are displayed by the bold line, with standard deviation shown with the shaded areas. Bound values for PBGNet$_\ell$ are trivial and thus not reported. We see that PBGNet bound minimization strikingly avoids overfitting by controlling the KL value according to the training set size. On the opposite, PBGNet$_\ell$ achieves lower test risk, but clearly overfits small training sets.

Table 4: Selected models overview.

| Dataset | Model | Hid. layers | Hid. size | $T$ | WD | C | KL | LR | Best epoch |
|---|---|---|---|---|---|---|---|---|---|
| ads | MLP | 3 | 100 | - | $10^{-4}$ | - | - | 0.1 | 9 |
| | PBGNet$_\ell$ | 1 | 10 | 100 | $10^{-6}$ | 11.44 | 14219 | 0.1 | 8 |
| | PBGNet$_{\ell\text{-bnd}}$ | 1 | 10 | 10 | $10^{-6}$ | 4.47 | 2440 | 0.001 | 101 |
| | PBGNet | 3 | 10 | 10000 | - | 0.47 | 27 | 0.01 | 49 |
| | PBGNet$_{pre}$ | 3 | 10 | 10000 | - | 0.58 | 0.09 | 0.1 | 82 |
| adult | MLP | 2 | 100 | - | 0 | - | - | 0.1 | 21 |
| | PBGNet$_\ell$ | 1 | 100 | 10000 | $10^{-6}$ | 2.27 | 13813 | 0.1 | 25 |
| | PBGNet$_{\ell\text{-bnd}}$ | 1 | 10 | 10 | 0 | 0.76 | 1294 | 0.001 | 111 |
| | PBGNet | 1 | 10 | 1000 | - | 0.30 | 226 | 0.1 | 78 |
| | PBGNet$_{pre}$ | 3 | 10 | 10000 | - | 0.09 | 0.13 | 0.01 | 73 |
| mnist17 | MLP | 2 | 50 | - | 0 | - | - | 0.01 | 56 |
| | PBGNet$_\ell$ | 3 | 10 | 100 | 0 | 19.99 | 5068371 | 0.1 | 15 |
| | PBGNet$_{\ell\text{-bnd}}$ | 1 | 10 | 10 | $10^{-6}$ | 2.82 | 690 | 0.001 | 86 |
| | PBGNet | 1 | 10 | 10000 | - | 1.33 | 164 | 0.1 | 106 |
| | PBGNet$_{pre}$ | 1 | 10 | 1000 | - | 0.73 | 0.46 | 0.1 | 71 |
| mnist49 | MLP | 2 | 100 | - | $10^{-6}$ | - | - | 0.001 | 32 |
| | PBGNet$_\ell$ | 2 | 50 | 10000 | 0 | 19.99 | 819585 | 0.01 | 33 |
| | PBGNet$_{\ell\text{-bnd}}$ | 1 | 10 | 10 | 0 | 2.40 | 1960 | 0.001 | 102 |
| | PBGNet | 1 | 10 | 10000 | - | 0.90 | 305 | 0.1 | 110 |
| | PBGNet$_{pre}$ | 1 | 100 | 10000 | - | 0.44 | 1.94 | 0.1 | 77 |
| mnist56 | MLP | 2 | 50 | - | $10^{-6}$ | - | - | 0.001 | 17 |
| | PBGNet$_\ell$ | 2 | 10 | 10000 | 0 | 19.99 | 883939 | 0.1 | 26 |
| | PBGNet$_{\ell\text{-bnd}}$ | 1 | 10 | 10 | 0 | 1.95 | 808 | 0.001 | 55 |
| | PBGNet | 1 | 10 | 10000 | - | 0.92 | 192 | 0.01 | 95 |
| | PBGNet$_{pre}$ | 1 | 50 | 10000 | - | 0.56 | 0.70 | 0.1 | 84 |
| mnistLH | MLP | 3 | 100 | - | $10^{-6}$ | - | - | 0.001 | 55 |
| | PBGNet$_\ell$ | 3 | 100 | 10000 | $10^{-6}$ | 19.99 | 98792960 | 0.1 | 92 |
| | PBGNet$_{\ell\text{-bnd}}$ | 1 | 10 | 100 | $10^{-6}$ | 2.00 | 8297 | 0.001 | 149 |
| | PBGNet | 1 | 10 | 50 | - | 0.81 | 1544 | 0.1 | 107 |
| | PBGNet$_{pre}$ | 2 | 100 | 10000 | - | 0.16 | 0.43 | 0.01 | 99 |

Table 5: Variance analysis of the experiment presented in Table 1. We repeated 20 times the experimental procedure described in Section 6, but with a fixed network architecture of a single hidden layer of 10 neurons to limit computation complexity. Each repetition is executed on different (random) train/test/valid dataset splits, and the stochastic gradient descent is initialized with different random weights. The resulting standard deviations highlight the stability of the models.

| Dataset | MLP | | PBGNet$_\ell$ | | PBGNet$_{\ell\text{-bnd}}$ | | | PBGNet | | | PBGNet$_{\text{pre}}$ | | |
|---|---|---|---|---|---|---|---|---|---|---|---|---|---|
| | $E_S$ | $E_T$ | $E_S$ | $E_T$ | $E_S$ | $E_T$ | Bnd | $E_S$ | $E_T$ | Bnd | $E_S$ | $E_T$ | Bnd |
| ads | 0.020 ± 0.005 | 0.034 ± 0.007 | 0.014 ± 0.003 | 0.029 ± 0.008 | 0.026 ± 0.003 | 0.035 ± 0.005 | 0.777 ± 0.000 | 0.112 ± 0.003 | 0.119 ± 0.007 | 0.218 ± 0.002 | 0.046 ± 0.007 | 0.048 ± 0.012 | 0.081 ± 0.007 |
| adult | 0.133 ± 0.007 | 0.149 ± 0.004 | 0.132 ± 0.003 | 0.148 ± 0.003 | 0.148 ± 0.002 | 0.151 ± 0.002 | 0.271 ± 0.002 | 0.156 ± 0.001 | 0.159 ± 0.002 | 0.215 ± 0.001 | 0.153 ± 0.003 | 0.154 ± 0.003 | 0.166 ± 0.002 |
| mnist17 | 0.002 ± 0.001 | 0.005 ± 0.001 | 0.002 ± 0.000 | 0.005 ± 0.001 | 0.004 ± 0.000 | 0.006 ± 0.001 | 0.102 ± 0.004 | 0.005 ± 0.000 | 0.006 ± 0.001 | 0.041 ± 0.001 | 0.005 ± 0.001 | 0.005 ± 0.001 | 0.010 ± 0.001 |
| mnist49 | 0.003 ± 0.002 | 0.013 ± 0.002 | 0.004 ± 0.001 | 0.016 ± 0.003 | 0.031 ± 0.001 | 0.033 ± 0.002 | 0.300 ± 0.000 | 0.039 ± 0.001 | 0.040 ± 0.003 | 0.143 ± 0.001 | 0.019 ± 0.002 | 0.019 ± 0.003 | 0.031 ± 0.002 |
| mnist56 | 0.004 ± 0.001 | 0.010 ± 0.002 | 0.003 ± 0.001 | 0.010 ± 0.002 | 0.020 ± 0.001 | 0.023 ± 0.002 | 0.186 ± 0.000 | 0.022 ± 0.001 | 0.023 ± 0.002 | 0.090 ± 0.001 | 0.012 ± 0.002 | 0.012 ± 0.003 | 0.020 ± 0.002 |
| mnistLH | 0.014 ± 0.003 | 0.032 ± 0.002 | 0.017 ± 0.001 | 0.038 ± 0.003 | 0.042 ± 0.001 | 0.049 ± 0.003 | 0.309 ± 0.001 | 0.054 ± 0.001 | 0.056 ± 0.002 | 0.162 ± 0.001 | 0.042 ± 0.002 | 0.042 ± 0.002 | 0.050 ± 0.002 |

Figure 7: Impact of the sample size $T$ on stochastic gradient descent solution test error for PBGNet$_\ell$ and PBGNet. Network parameters were fixed with a single hidden layer of size 10 and trained with initial learning rate of 0.1. For each sample size values and the combinatorial sum approach, 20 repetitions of the learning procedure with different random seeds were performed: each of them executed on different (random) train/test/valid dataset splits, and the stochastic gradient descent is initialized with different random weights. The test error mean of the learned models is displayed by the bold line, with standard deviation shown with the shaded areas.

## Footnotes

[4]See code in the following file: https://github.com/gletarte/dichotomize-and-generalize/blob/master/pbgdeep/networks.py.