[Reviews · NeurIPS 2019]

Reviewer 1



The main contribution of the paper is to devise a bound that only requires data to be i.i.d. with the condition that the activation functions are sign functions. This contrasts with previous approaches who had non-obvious assumptions on e.g. the margin. I'm not an expert in PAC-Bayes but this definitely looks like a step in the right direction. The authors do a good job of providing the relevant background about general PAC-Bayes and the application to linear classifiers, which provide the backbone of the paper. The proposed learning method is arguably not that useful for practitioners given the computational complexity and the performance, but it provides a good testbed for the theory. Given this, I would prefer to see more space devoted to discussing the implications of the theory rather than to the experimental results. Table 1 should contain some type of error bars, given the stochastic nature of the optimization. In fact, the authors state in the reproducibility checklist that they provide error bars, so this should be addressed. Nits: * The citation in line 137 feels a bit weird, since SGD has been the standard approach for learning deep networks for much longer.o * Few typos (e.g. line 163 "leaning" -> "learning")

Reviewer 2



This work considers the multilayer neural networks with binary activation based on PAC-Bayesian theory. The main contributions include: 1) an end-to-end framework to train a binary activated deep neural network and 2) nonvacuous PAC-Bayesian generalization bounds for binary activated deep neural networks.Some empirical studies are provide to verify the effectiveness of the proposed algorithm. This is a theoretical submission, and I have some concerns as follows: 1) This work focuses on binary activation in neural network, while the authors does not present some real applications, as well as some motivations of such activation. As I know, most common-used activation functions are continous and lipschitz, such as sigmoid and Relu function. 2) In real applications, how to decide the prior parameter \mu, especially for deep neural network. This is very important to use the relevant theory. From a theoretical review, it is quite straight to obtain the theorem from the traditional PAC Bayes analysis wihout technical new insights. 3) The experimental results do not well support the theoretical analysis. There is no necessary comparisons with state-of-the-art algorithm on deep learning, and I do not find some new insights in experiments.

Reviewer 3



# Description The paper has two complementary contributions. One contribution is a new approach for training neural networks with binary activations. The second contribution is PAC-Bayesian generalization bounds for binary activated neural networks that, when used as the training objective, come very close to test accuracy (i.e. very much non-vacuous). # Strength From the experiments it is clear that optimizing the structural risk results in much better generalization bounds than optimizing empirical risk. The gap between the training and test performance is also much smaller. I think this is very promising for training more robust networks. The method actually recovers variational Bayesian learning when the coefficient C is fixed, but in contrast to it, this coefficient is learned in a principled way. # Discussion The claimed results and experiments look great. However, several things are not fitting together. The paper claims to provide non-vacuous generalization bound for binary activated networks. First, I do not clearly see where the specialization to binary activations is theoretically allowing or improving the bound? The links seems to be through the use of approximation with a tree, derived for binary networks and substituting its KL divergence into the known bound. The coefficients in the KL divergence of the tree approximation (17) are growing combinatorially and certainly do not make the bound tighter. Second, the bound holds for the expected prediction with random weights and not deterministic binary weights. This expectation is different in the original network and in the tree-structured computation graph used for the approximation and the bound. Therefore, if we speak of a theoretically provable bound, there is strictly speaking, no such bound shown for binary activated networks. Despite contrasting to the literature (in line 16, where it is said that the stochastic network is not the one used in practice) in the proposed method the network is also stochastic. So one has to use averaged predictor in practice or the bound is not valid. The approximation with the tree structured computation, made the training more tractable, but computing the expectation still requires either 2^d complexity per layer or sampling similar to that in the BAM. The results inferred for the case of one hidden layer, contain detailed expressions, but I believe match to the known higher level results. The model with noises on the parameters defines conditional probabilities p(s|x) and p(y|s), where y = sgn(w2 s). The model for p(y|x) is thus of the structure of a Bayesian network with {-1,1} states. More specifically, it is similar to sigmoid belief network (Neal 1992), where the standard logistic noise is added to pre-activation rather than the Gaussian noise added in front of the linear mapping of parameters. In either case, one can obtain general expressions for the expected value and derivatives (12-13) in terms of these conditional probabilities and their derivatives. Furthermore, the MC approximation (14)-(15), is known as REINFORCE or score function estimator. I believe it would be clearer to rewrite section 3, in this more general setting. Note that there is a broad literature on estimating gradients in stochastic neural networks. The REINFORCE method readily extends to deep networks, suggesting that the tree-structured approach in section 4 may be not needed. However, it may have a very high variance in practice. Recent variance reduction techniques with this method include: Yin and Zhou, ARM: Augment-REINFORCE-Merge Gradient for Stochastic Binary Networks. Tucker et al., REBAR: REBAR: Low-variance, unbiased gradient estimates for discrete latent variable models. Grathwohl et al., Backpropagation through the Void: Optimizing control variates for black-box gradient estimation. It would be interesting to include a discussion on the connection of (5) with variational approach to Bayesian learning. # Experiments It is somewhat unsatisfactory that the estimate of the true risk, which is the empirical measure of generalization, is still better for the methods that have vacuous theoretical bounds than for the proposed method that has very tight theoretical bounds. How much the error rates and bounds are repeatable over random initializations and splits of the data into train-val-test sets? I think that the MC sample is not very crucial during training (as studied in Fig. 5). The expectation over training data and expectation over samples form one joint expectation and then it is up to dynamics of SGD how well this will be optimized. Studying dependance on the training sample size could be more interesting, to see the power of regularization in the case of small training sample size. # Minor 37-43 In my opinion the outline is not very useful in small papers, I usually skip it. I think this space can be better used e.g. for a figure from supplementary. # Final Comments In the rebuttal authors promised to address all concerns I raised, improve clarity and experiments, which I am fully satisfied with. With these improvements it will make a high quality paper. One question that remains unclear to me, is whether the bound is new and whether binary activations are essential for the bound. From the comment by Reviewer 2: "From a theoretical review, it is quite straight to obtain the theorem from the traditional PAC Bayes analysis without technical new insights." I conclude it is not. But the authors insist it is novel and has some features specific to binary activations. "Relying on binary activation function allows us to express a close-formed solution for the PAC-Bayes bound, without other assumptions than the iid one. Another appeal of our bound: it relies more on the network architecture (e.g., dk, the layer width of each layer k, appears in Eq. 16) than previous results (in the seminal work of Dziugaite and Roy, 2017, a posterior distribution is used over a set of a neural network weights, without taking into account any other architecture specific information)" I ask the authors to make this clear in the final version. The specific architecture dependent bound appears only in the case when the tree-inference is used. If the the approximation is not used, but we still consider binary activated networks, no new specific bound arises. Furthermore, the bound with the tree inference is looser than without. This is an indication that the the tree inference has a higher complexity and is perhaps inferior.

[Author Response · NeurIPS 2019]

We would like to thank all three reviewers for their constructive assessment of our work.

**Reviewer 1**

*Importance of the experimental results:* Non-vacuous generalization bounds are arguably desirable for delicate machine
learning applications. Since such results are actually rare for neural networks, we consider important to show empirically
that our approach leads to sound performances. We believe that this can encourage others to develop similar methods
that will eventually lead to higher practical impact, following this work.

*Error bars:* The reviewer concern is *partially* addressed in Section B.3 of the appendix, through the Monte Carlo
sampling size effect study. Indeed, Figure 5 displays the test error — with error bars — for PBGNet and PBGNet$_\ell$
when varying the sample size. Note that each result is obtained by averaging over 20 repetitions of the learning
procedure, each of them executed on different (random) train/test/valid dataset splits, and the stochastic gradient descent
is initialized with different random weights. That being said, we undertake to push forward the variance analysis for the
final version of the paper, as detailed in Reviewer 3 *Experiments* section.

**Reviewer 2**

*Choice of the activation function*: A variety of activation functions exists in the literature and we obviously do not focus
on the most common one. Nevertheless, the sign activation function is used in the binary networks cited in the paper,
notably to reduce the memory footprint of such networks which could be embedded on small devices (e.g., Bengio,
2009). In our context, the sign activation is crucial to apply the mathematical trick, and express the predicted outcome
as the erf function. This gives a principled way of training the network and derive the PAC-Bayes generalization bound.

*Choice of prior*: The bound holds regardless of the choice of the prior $\mu$. In our experiments, we centered the prior on
the SGD initialisation weights (as in Dziugaite and Roy, 2017), which corresponds to the real-life scenario where one
does not have prior knowledge about the task at hand. The PBGNet$_{\text{pre}}$ variant of our algorithm opens the way for using
a prior from a precedent learning task, as the transfer learning scenario mentioned in the paper (see Lines 219-222).

*Mismatch between theory and experiments*: We worked hard to provide a rigorous and honest empirical study of our
theoretical analysis strengths and weaknesses. It is certainly disappointing that there exist good predictors with trivial
bounds (as mentioned by Rev. 3), but we still managed to obtain very tight bounds for more than decent predictors.
Note also that we compared to tanh networks as this activation is similar to the erf function derived from our analysis.
It allows us to use the same optimisation scheme and hyperparameter grid to compare the methods on equivalent basis.

**Reviewer 3**

*Improvement due to the binary activations:* Relying on binary activation function allows us to express a close-formed
solution for the PAC-Bayes bound, without other assumptions than the *iid* one. Another appeal of our bound: it relies
more on the network architecture (*e.g.,* $d_k$, the layer width of each layer $k$, appears in Eq. 16) than previous results (in
the seminal work of Dziugaite and Roy, 2017, a posterior distribution is used over a set of a neural network weights,
without taking into account any other architecture specific information).

*Bounds for (a unique) binary activated networks:* The reviewer is right to say that our analysis does not provide
guarantees for a single deterministic binary activated (BAM) network, but for a continuous aggregation of such BAM
networks. We clearly mention this in the introduction (Lines 26-27), but we agree that Line 16 may be ambiguous
and we will rephrase it. That being said, several points are worth mentioning: (i) Even if computationally expensive,
our predictor closed-form expression is deterministic; (ii) The prediction using Monte Carlo sampling empirically
shows a small standard deviation consistently below $10^{-3}$, as discussed in the appendix (see Lines 432-448); (iii) In our
experiments, we observed that predicting with the single *Maximum-A-Posteriori* BAM network generally gives results
remarkably close to the aggregated PBGNet predictor (hinting that the posterior may be quite peaked). Recall that this
unique BAM network represents exactly the same prediction function as its mapped tree predictor. This preliminary
observation suggests that the bound can provide an appropriate guide to train a BAM network.

*Link with similar optimization procedures:* There is definitely a strong connection between our optimization procedure
and the REINFORCE method. We greatly thank the reviewer for pointing us the relevant literature; we were genuinely
not aware of it. Indeed, we will rewrite parts of Section 3 to highlight these connections, and express our sampling
scheme as a particular case of a general technique rather than a new one. As a matter of fact, we think this will enrich the
paper, as our PAC-Bayesian analysis support existing strategies to train non-differentiable neural networks. Moreover,
the connection with Variational Bayes (for a fixed $C$) is also very relevant, and will be mentioned in the paper.

*Experiments:* We share the reviewer concerns about the lack of standard deviation analysis in Table 1. Given our
computing resources, we cannot provide these results in this rebuttal. Nevertheless, we commit to produce for the final
version of the paper a thorough stability analysis with 20 different random train/test/valid splits for all six datasets and
five considered models of Table 1, for a slightly reduced hyperparameters search grid. Furthermore, we will add a study
of the training sample size effect on PBGNet$_\ell$ and PBGNet with fixed parameters for the biggest dataset (mnistLH).

[Meta-Review · NeurIPS 2019]

This work studies PAC-Bayes bound optimization in the setting of deep neural networks with binary activations. One of the stated contributions of the paper---showing how to optimize despite the binary activations providing no naive derivative---is, in fact, a known technique in the literature on variational inference. This somewhat undermines the impact of the work, though importing these ideas into the PAC-Bayes community is nice. The other contribution is obtaining nonvacuous bounds and here it is impressive to see such tight bounds. I have a few issues to raise with the introduction, which I would like addressed in revisions: First, the authors write: "Although informative, these results upper bound the prediction error of a (stochastic) neural network with perturbed weights, which is not the one used to predict in practice". I find this statement somewhat odd because, as far as I can tell, the present paper also doesn't resolve the gap between "the networks used in practice" and "the predictors for which the bounds hold". A review points this out too. Yes, one obtains bounds for the weighted vote, not the Gibbs classifier, but people also don't use weighted votes in practice. (Though, they may start based on evidence that these help against adversaries.) Second, the authors refer to Neyshabur et al as work that presents bounds that depend on the architecture. Indeed they do, but their approach to obtain a bound on the deterministic classifier produces a _completely_ vacuous bound for standard networks. The technique is potentially important, but, at present, the bound does not explain anything. So based on the aesthetic principles of this work, it seems odd to me to bury this issue and to suggest that "spectral norms tell us something valuable about generalization". Do they? Unlikely.